# PixFoundation: Are We Heading in the Right Direction with Pixel-level Vision Foundation Models?

## Abstract

Multiple works have emerged to push the boundaries on multi-modal large language models (MLLMs) towards pixel-level understanding. The current trend in pixel-level MLLMs is to train with pixel-level grounding supervision on large-scale labelled data with specialized decoders for the segmentation task. However, we show that such MLLMs when evaluated on recent challenging vision-centric benchmarks, exhibit a weak ability in visual question answering (VQA). Surprisingly, some of these methods even downgrade the grounding ability of MLLMs that were never trained with such pixel-level supervision. In this work, we propose two novel challenging benchmarks with paired evaluation for both VQA and grounding. We show that MLLMs without pixel-level grounding supervision can outperform the state of the art in such tasks. Our paired benchmarks and evaluation enable additional analysis on the reasons for failure with respect to VQA and/or grounding. Furthermore, we propose simple baselines to extract the grounding information that can be plugged into any MLLM, which we call PixFoundation. More importantly, we study the research question of "When does grounding emerge in MLLMs that are not trained with pixel-level grounding supervision?" We show that grounding can coincide with object parts, its location, appearance, context or state, where we show 27-45% of the examples in both benchmarks exhibit this phenomenon. Our code and datasets will be made publicly available and some are in the supplemental.

## 1 Introduction

There have been numerous advancements in pixel-level image and video understanding, including tasks such as image/video segmentation Zhou et al. (2022); Minaee et al. (2021); Kirillov et al. (2023); Ravi et al. (2024), pixel-level visual grounding and reasoning Rasheed et al. (2024); Lai et al. (2024), depth estimation Yang et al. (2024) and tracking Wang et al. (2023). The majority of these have been transformed with the emergence of foundation models Bommasani et al. (2021), specifically multi-modal large language models (MLLMs) Liu et al. (2023/); Dai et al. (2023). Nonetheless, pixel-level MLLMs have shown degradation in their capabilities and chat performance Lai et al. (2024). Recent models tried to address this gap Zhang et al. (2024b,a), yet they relied on standard evaluation benchmarks, overlooking the shortcomings of current MLLMs.

Recent efforts explored the shortcomings of MLLMs in vision-centric benchmarks Tong et al. (2024b,a). Such benchmarks focused on challenging visual tasks such as counting. Nonetheless, these benchmarks did not evaluate the recent pixel-level MLLMs and rather used the visual question answering task as a proxy to evaluate MLLMs' grounding ability. In this work, we propose challenging vision-centric benchmarks that are dedicated to evaluating pixel-level MLLMs and provide a comprehensive paired evaluation for both VQA and grounding, which we call PixMMVP and PixCV-Bench. Our paired evaluation means that the referring segmentation is related to the object of interest in the visual question, providing a better analysis of MLLMs' capabilities. Through

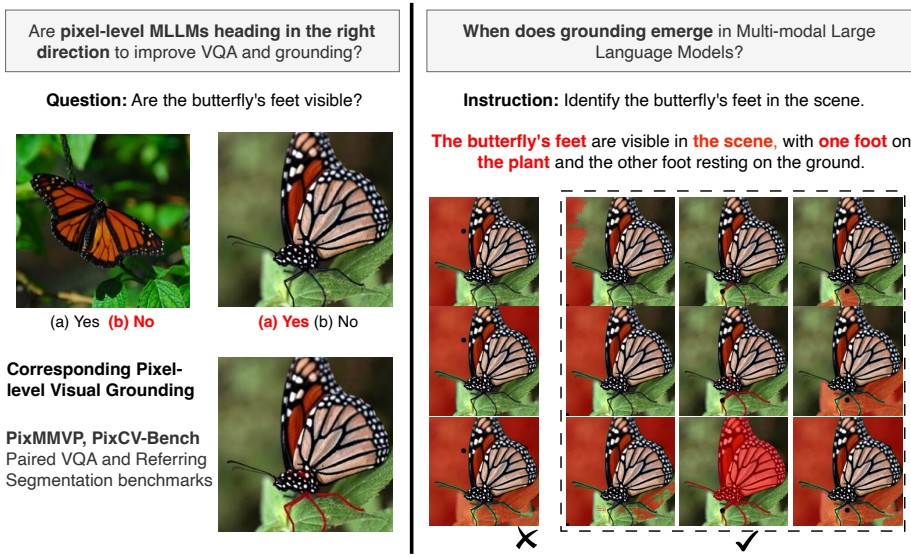

Figure 1: The two major research questions we explore: (i) the grounding & VQA ability of pixel-level MLLMs in challenging scenarios (left), (ii) the ability of vanilla MLLMs to perform grounding and when does it emerge (right). **Right:** shows the noun phrases and their corresponding predicted segmentation, highlighted in red, extracted from LLaVA 1.5 attention maps with three masks due to point prompt ambiguity from the maximum attention, highlighted as a black circle. Note that not all noun phrases and segmentations are shown for space constraints.

these, we answer the first research question; *"Are the current pixel-level MLLMs trained with full grounding supervision heading in the right direction to improve both grounding and visual question answering (VQA)?"*. Our findings show that the majority of pixel-level MLLMs still fall short in such a challenging setting. While evidently, some of these show superior performance in visual grounding, we show that MLLMs that were not trained with pixel-level grounding and without using specialized segmentation decoders can have better performance.

There have been recent works showing training-free segmentation emerging from vision language models Wang et al. (2024); Luo et al. (2024); Hajimiri et al. (2025). Concurrent work has specifically explored emerging grounding in MLLMs Cao et al. (2024). Another concurrent work Wu et al. (2024) has observed the degradation of pixel-level MLLMs' VQA abilities. Nonetheless, previous efforts used standard evaluation benchmarks that evaluate each task separately. Our benchmarks provide a paired VQA and referring segmentation evaluation, where we propose an evaluation metric that takes into account the performance in both. Such paired benchmarks not only provide better scoring for pixel-level MLLMs performance, but they are designed to be vision-centric, with a focus on what MLLMs fall short in. Moreover, they provide the means to interpret the failures of these MLLMs and whether they are stemming from grounding, VQA or both. More importantly, unlike concurrent efforts, we focus on the second research question of *"When does grounding emerge in MLLMs that are not trained with pixel-level supervision?"*. Our work documents that emerging grounding in MLLMs does not necessarily coincide with the exact language tokens of the object, as shown in Fig. 1. We show that up to 45% and 27% of the examples in PixMMVP and PixCV-Bench, respectively, have grounding coinciding with concepts about the referred objects' parts, position, color or context.

In summary, our contributions include: (i) Proposing paired pixel-level vision-centric benchmarks, PixMMVP and PixCV-Bench, with segmentation annotations and referring expression of the object of interest in the corresponding questions. (ii) Benchmarking recent efforts in pixel-level MLLMs where we show that they degrade VQA capabilities. More importantly, some of them lag in visual grounding with respect to simple techniques of extracting the segmentation from vanilla MLLMs, i.e., MLLMs that are not trained for pixel-level grounding. (iii) We provide a simple mechanism for extracting segmentation from vanilla MLLMs, with an understanding of when grounding emerges, that surpasses the state of the art. Our mechanism uses the observation that grounding can emerge corresponding to different output tokens describing the object's appearance or location, not necessarily the exact text of the object of interest, which we call PixFoundation.

## 2    Related work

**Pixel-level vision foundation models.** There have been various vision foundation models trained for the segmentation task (e.g., SAM and SAM 2.0) Kirillov et al. (2023); Ravi et al. (2024). Orthogonal to this, some methods discussed the ability of vision foundation models such as CLIP and BLIP in image segmentation without any segmentation supervision Luo et al. (2024); Hajimiri et al. (2025); Wang et al. (2024). Yet, they relied on earlier foundation models that did not incorporate the power of large language models. Combining large language models with vision has been extensively researched with pioneering works such as LLaVA Liu et al. (2023/, 2024) and instruct-BLIP Dai et al. (2023). Multiple works afterwards focused on pixel-level visual grounding in these MLLMs with full supervision and specialized segmentation decoders Lai et al. (2024); Rasheed et al. (2024); Zhang et al. (2024a,b,a,b). However, these methods were lagging in their chat performance. Notably, pixel-level MLLMs were not evaluated on the challenging benchmarks that focused on the shortcomings of MLLMs Tong et al. (2024b,a). Hence, it is still unclear if the pixel-level grounding supervision helped to improve their ability on these challenging tasks or not. In this work, we focus on the previous question to have a better understanding of their performance. Concurrent work has shown that without pixel-level supervision, there is an emerging ability to perform pixel-level grounding Cao et al. (2024). We rely on this method as our baseline, but unlike previous works, we provide an insight into when grounding emerges in such MLLMs. We propose a baseline that uses a novel and simple mechanism to perform mask selection while taking the previous insight into consideration.

**Benchmarking multi-modal large language models.** There is an abundance of standard benchmarks used for evaluating MLLMs (e.g., MMU Yue et al. (2024)) and pixel-level benchmarks (e.g., refCOCO/+/g Yu et al. (2016); Kazemzadeh et al. (2014)). These have pushed the limits on MLLMs capabilities in terms of VQA and visual grounding. Nonetheless, there have been various works that discussed the shortcomings of MLLMs. One of them discussed the shortcomings in CLIP Radford et al. (2021), which is used in various MLLMs as a visual backbone. They proposed a benchmark, MMVP Tong et al. (2024b), that is focused on the visual aspects within a VQA task. More recently, CV-Bench Tong et al. (2024a) focused on two major tasks that are vision focused which are counting and relative positioning. Both were proposed to evaluate MLLMs that do not have the ability to generate segmentation output. Nonetheless, they still provide quite challenging scenarios that can act as a strong benchmark for the pixel-level MLLMs counterpart. In this work, we extend these two benchmarks with pixel-level annotations and referring expressions that correspond to the object of interest within the VQA task, and propose a paired evaluation metric.

## 3    Method and benchmarks

In this section, we describe our two benchmarks and probing techniques for pixel-level MLLMs and MLLMs that were not trained with pixel-level grounding supervision.

### 3.1    Paired Benchmarks for VQA and Grounding

**PixMMVP benchmark:** We build upon the recently released MMVP Tong et al. (2024b) which identified clip blind pairs and used them to build a challenging benchmark with the corresponding questions and choices for 300 images. We manually annotate each question with the corresponding object of interest referring expression, e.g. an elderly person or the butterfly's feet. There are seven questions only that are not designed to inquire about a specific object in the scene, which are excluded, such as questions inquiring on the view direction of the camera. The referring expressions in our dataset correspond to what needs to be grounded in the image to answer the question and are as fine-grained as possible. Afterwards, we manually label these objects of interest with polygonal annotations using the VGG annotator Dutta et al. (2016). Hence, we create the first paired benchmark for both VQA and pixel-level visual grounding.

**PixCV-Bench benchmark:** For this benchmark we build upon the 2D component of the recently released CV-Bench Tong et al. (2024a). We specifically select the 2D component, since they are sourced from segmentation datasets (i.e., ADE20K Zhou et al. (2017) and COCO Lin et al. (2014)), which can be used in our proposed benchmark. However, the publicly released CV-Bench does not identify the objects in question and their corresponding segmentation. As such we use GPT-4o to parse the questions and identify the objects of interest automatically, followed by manual inspection

and correction. Specifically, we collect the classes in each image from the corresponding dataset and construct a list of class choices "1. <CLS1>, 2. <CLS2>, ...". Then we prompt GPT-4o with the following, *"Provide number only as an answer. Identify the objects of interest in the following question: <QUESTION> ? 1. <CLS1>, 2. <CLS2>, ... "*. This provides us with the categories per question that highlight the objects of interest. While seemingly these are categorical annotations, not referring expressions, certain scenarios in CV-Bench are different. Specifically, in the relative positioning task, all the questions that include an object highlighted by a red box in the image are annotated with the referring expression, "(annotated by the red box)", beyond simple categories.

Afterwards, we use the selected categories from GPT-4o to retrieve the corresponding segmentation mask per image. Furthermore, we use a custom annotation tool to manually filter the objects in the question, e.g. selecting only the object mask annotated by the red box and filtering out other instances. Another example that needs manual filtration, when the class in question is a broader category than what is inquired upon, e.g., "Pendant Lamp" which is under the category of "Lamp" in ADE20K. In such a case, we filter out the masks of other types such as "Table Lamp". Moreover, we identify missing annotations and manually annotate these missing objects. We provide the final paired PixCV-Bench with referring expressions, their segmentation annotations, visual questions and corresponding answers that can be used to evaluate the grounding ability in relation to the original VQA task. Appendix A provides visual examples from our benchmarks.

## 3.2 A Pixel-level MLLMs study

We utilize the two proposed benchmarks, PixMMVP and PixCV-Bench, to evaluate the current trend in pixel-level MLLMs that rely on pixel-level supervision and specialized segmentation decoders. Furthermore, we inspect the failures of these pixel-level MLLMs and explore simple approaches to pixel-level understanding from MLLMs that overcome the previous shortcomings.

**Pixel-level MLLMs shortcomings.** We highlight the failures of the current state-of-the-art pixel-level MLLMs through three probing techniques. First, we highlight the degraded performance in VQA from most of these MLLMs that are trained with pixel-level supervision. We use for that the following prompt, *"<QUESTION>? <OPTION1> <OPTION2>..."*, as shown in Figure 2a. Notably, the worst two models in this task, LISA Lai et al. (2024) and GLAMM Rasheed et al. (2024), are not able to provide an answer and rather refer to a segmentation mask. On the other hand, OMG-LLaVA Zhang et al. (2024b) shows better ability in VQA.

The second shortcoming we discuss is that these MLLMs exhibit a degraded ability to follow instructions. In order to probe this, we use the following prompt: *"<QUESTION>? a.<OPTION1> b.<OPTION2>... Answer with the option's letter from the given."* Figure 2b shows an example with the answers from the worst two models in this aspect which are LISA Lai et al. (2024) and LLaVA-G Zhang et al. (2024a). Both are incapable of following the instruction, yet LLaVA-G tries to tackle the question, unlike LISA. On the other hand, OMG-LLaVA shows better ability to follow the instructions and answer the questions.

Third, we highlight their degraded ability to visually ground objects. Surprisingly, although they were trained with pixel-level grounding supervision, not all of these models show superior grounding performance. Figure 2c shows the second prompt to generate a segmentation mask for the ground-truth referring expression. The purpose of this probing is to understand whether the failure in these models is purely on the VQA task, or its inability to ground the objects of interest in the corresponding question or both. Figure 2c shows the worst two models in this aspect, which are GLAMM, the region captioning variant, and LLaVA-G. Both fail to segment the specific object in question, while OMG-LLaVA shows better performance.

**Baselines and upper bounds.** In addition to evaluating state-of-the-art pixel-level MLLMs, we propose two baselines and one upper bound. The first of which is inspired by a concurrent work Cao et al. (2024) that identified the emergent grounding in multi-modal large language models without the need for any pixel-level grounding supervision. Specifically, we use their attend and segment meta architecture as one of our baselines. However, we are the first to discuss when such grounding emerges in these models. We identify an interesting connection between the identified output tokens and the output grounding from the attention maps that gives insights into how these models reason.

The attend and segment meta-architecture extracts the raw attention map for the $i^{th}$ output token, $A_i \in [0,1]^{n_{\text{layer}} \times n_{\text{head}} \times (x+hw+y+i-1)}$, where $n_{\text{layer}}, n_{\text{head}}$ are the number of layers and heads, resp.

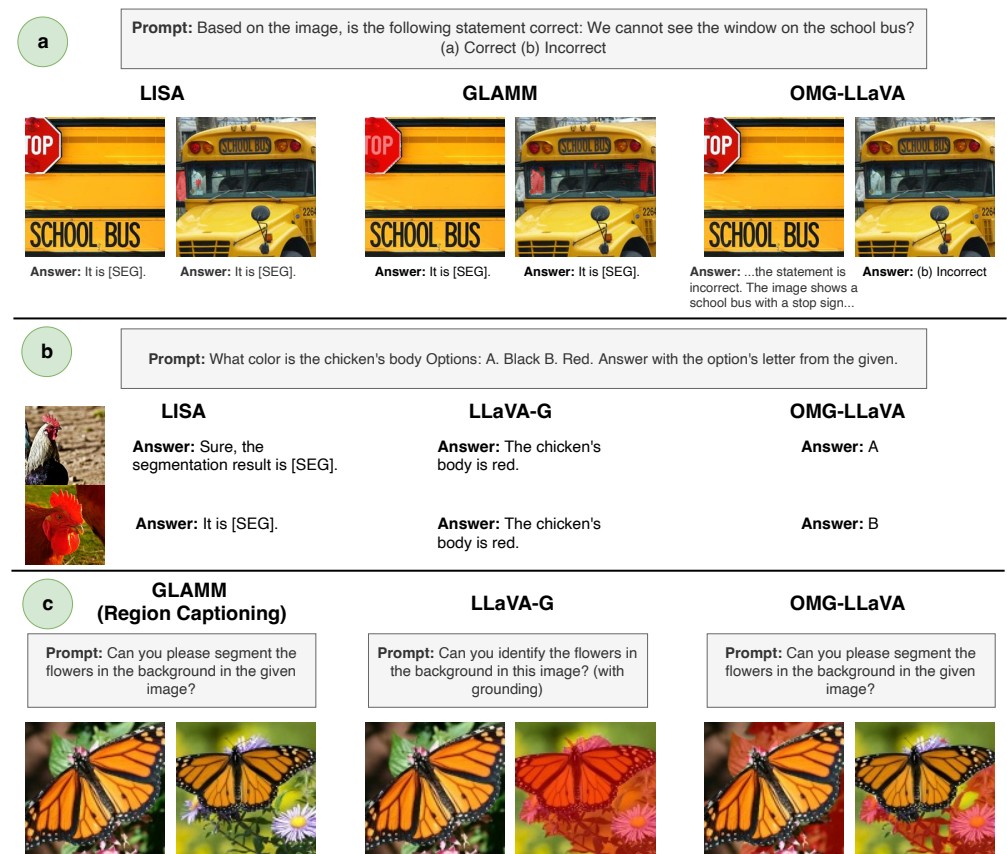

Figure 2: Shortcomings of pixel-level MLLMs. (a) The first shortcoming of pixel-level MLLMs is the degraded performance in visual question answering. (b) The second shortcoming of pixel-level MLLMs, which relates to the first, is the degraded performance in instruction following, where the question is instructing the model to generate one letter from the options. Even when the model tries to answer the question it still fails to properly answer with one option letter. (c) The third shortcoming of pixel-level MLLMs is the degraded performance in pixel-level visual grounding in certain models. The predicted segmentation masks corresponding to the [SEG] token/s are highlighted in red.

Then, $x, y$ are the number of input language tokens before and after the visual tokens, respectively, while $hw$ are the height and width of the input image. Only the attention corresponding to the visual tokens of length $hw$ is used, and these attention maps are averaged across the layers and heads, resulting in $\bar{A}_i \in [0, 1]^{h \times w}$. This is further normalized across all the output, $\tilde{A}_i = \bar{A}_i - \frac{1}{N} \sum_{j=1}^{N} \bar{A}_j$ for $N$ output tokens. The attend and segment depends on using the spaCy natural language processing tool Honnibal et al. (2020) to identify the noun phrases and associate them with the ground-truth referring expressions. Thus, the spaCy embeddings closest to the ground-truth expression are used in the mask selection. This is followed by extracting the maximum attention point to feed into SAM Kirillov et al. (2023) as a point prompt.

For our baseline and upper bound, we build upon the previous pipeline and build an *oracle* upper bound and an *automatic* baseline. We introduce two main modifications to account for our observation that the correct grounding can occur with different output tokens describing the object, not necessarily aligning with the exact ground-truth expression. The first modification is to inspect all the output tokens without relying on spaCy embeddings. In the *oracle* we rely on the ground-truth mask to select the correct token and its corresponding segmentation with the highest intersection over union as an upper bound. The *automatic* baseline uses a simple but powerful mechanism where we highlight the predicted masks on the original image to show the potential object of interest. This is followed by feeding these images to a multi-modal LLM to inquire which is best in highlighting this object.

| Image | Referring Expression | Concept Category | Noun Phrase | Output |
|---|---|---|---|---|
| 1 | the butterfly's wings | Color & Appearance | orange wings | In the image, there is a butterfly with orange wings. |
| 3 | the flame of the match | Location & Position | the top | The flame of the match is located at the top of the image, surrounded by darkness. |
| 6 | the dog's face | Color & Appearance | a black and white dog | The dog's face in the scene is a black and white dog with a black nose. |
| 161 | the minute hand of the clock | Location & Position | the 12 o'clock position | The minute hand of the clock in the scene is located at the 12 o'clock position. |

| 1 | 3 | 6 | 161 | Prompts Ablation |

Figure 3: Examples of concept categories where the grounding emerges in PixMMVP using LLaVA 1.5 (7B). **Top:** referring expression, output response, noun phrases and concepts corresponding to the grounding using the *oracle* selection. **Bottom:** the four images with predicted segmentation mask, highlighted in red, using the *oracle* selection. The input point prompt highlighted as a black circle. It shows the segmentation of the referring expression emerging in different output noun phrases than the original expression. The final plot at the bottom shows the ablation on the different input prompts to SAM using a random input point vs. the maximum attention point (First) vs. the second vs. the third maximum, paired with our *oracle* selection. $\mathcal{M}$: mean intersection over union.

Specifically, we use the following prompt *"Select the image that has <EXPR> best highlighted in red color than the others? Answer with a number from 1 to <N> and mention the number only. "*, where <EXPR> and  are the ground-truth expression and the image tokens respectively. In our *automatic* baseline, we rely on GPT-4o for the mask selection, refer to the App. E for the mask selection results using the open source Cambrian (8B). The second modification, since SAM has a good understanding of point prompting ambiguity, we process three potential masks for each prompt instead of one. This enables us to utilize the power of SAM in identifying fine-grained objects and referring expressions that tend to surpass what other MLLMs do, even those trained with pixel-level grounding supervision. Figure 3 shows qualitative results where the segmentation emerges, corresponding to output tokens describing the object in terms of color or location instead of the exact ground-truth referring expression, motivating our *oracle* and *automatic* baseline. Interestingly, our *oracle* enables a quantifiable study of this phenomenon that can better interpret these MLLMs.

## 4 Experiments

### 4.1 Experimental Setup

**Evaluation benchmarks, protocols and metrics.** PixMMVP is composed of 300 images paired with questions, choices, referring expressions and segmentation masks, while PixCV-Bench has 1,438 images with their corresponding annotations similarly. On each benchmark, we evaluate the VQA and visual grounding capabilities following three probing techniques and report their metrics. The first probing is to evaluate the VQA ability, where the accuracy is computed using GPT-4o following Tong et al. (2024b) as, $\mathcal{A}\dagger$. If the model generates a segmentation without explicitly asking it to, it is evaluated with respect to the ground-truth referring segmentation in terms of mean intersection over union as $\mathcal{M}\dagger$. The second probing prompts the model to identify the referred expression then evaluates the mean intersection over union reported as $\mathcal{M}$. The third probing following Tong et al. (2024a) instructs the model to generate a single option letter and evaluate the accuracy directly without GPT-4o, reported as $\mathcal{A}$. There is a need for the first probing since some of the recent pixel-level MLLMs face challenges in following instructions. We evaluate the score of each model, $\mathcal{S}$, which is

| Method | PixGr. | PixMMVP | | | | | PixCV-Bench | | | | |
|---|---|---|---|---|---|---|---|---|---|---|---|
| | | $\mathcal{A}$† | $\mathcal{A}$ | $\mathcal{M}$† | $\mathcal{M}$ | $\mathcal{S}$ | $\mathcal{A}$† | $\mathcal{A}$ | $\mathcal{M}$† | $\mathcal{M}$ | $\mathcal{S}$ |
| LLaVA 1.5 (7B) | ✗ | 27.3 | 28.0 | - | - | - | 17.4 | 60.3 | - | - | - |
| LLaVA 1.5 (13B) | ✗ | 39.3 | 30 | - | - | - | 14.5 | 61.4 | - | - | - |
| Cambrian (8B)* | ✗ | 52.0 | 52.0 | - | - | - | 62.2 | 72.2 | - | - | - |
| OMG LLaVA (7B)** | ✓ | 12.0 | 12.0 | 17.8 | 38.0 | 18.2 | 12.0 | 42.1 | - | 50.5 | **45.9** |
| GLAMM (7B) | ✓ | 1.3 | 2.7 | 31.5 | 47.4 | 5.1 | - | - | 30.2 | 51.9 | - |
| GLAMM - RegCap (7B) | ✓ | 12.7 | 6.7 | 14.5 | 18.6 | 15.1 | 27.8 | 54.4 | 3.6 | 7.4 | 13.0 |
| LISA (7B) | ✓ | 7.3 | - | 18.1 | 42.9 | 12.5 | 3.7 | - | 16.8 | 48.1 | 6.7 |
| LLaVA-G (7B) | ✓ | 9.3 | - | 17.8 | 13.5 | 12.2 | 14.1 | 4.4 | 1.7 | 17.6 | 15.8 |
| LLaVA 1.5 (7B) + (a+s) | ✗ | 27.3 | 28.0 | 11.1 | 11.2 | 16.0 | 17.4 | 60.3 | 5.2 | 15.7 | 24.9 |
| LLaVA 1.5 (13B) + (a+s) | ✗ | 39.3 | 30 | 9.8 | 11.4 | 17.7 | 14.5 | 61.4 | 4.7 | 14.9 | 24.0 |
| Cambrian (8B)* + (a+s) | ✗ | 52.0 | 52.0 | 14.3 | 15.1 | 23.4 | 62.2 | 72.2 | 18.6 | 15.9 | 29.6 |
| PixFoundation (7B) (Ours) | ✗ | 27.3 | 28.0 | 18.8 | 25.9 | 26.9 | 17.4 | 60.3 | 5.4 | 28.5 | 38.7 |
| PixFoundation (13B) (Ours) | ✗ | 39.3 | 30 | 16.9 | 25.0 | 30.6 | 14.5 | 61.4 | 4.8 | 27.6 | 38.1 |
| PixFoundation (8B)* (Ours) | ✗ | 52.0 | 52.0 | 29.6 | 30.3 | **38.3** | 62.2 | 72.2 | 23.9 | 33.1 | 45.4 |
| **Upper Bound - Oracle Selection** | | | | | | | | | | | |
| PixFoundation† (7B) (Ours) | ✗ | 27.3 | 28.0 | **26.1** | **38.0** | **32.2** | 17.4 | 60.3 | **6.3** | **49.7** | **54.5** |
| PixFoundation† (13B) (Ours) | ✗ | 39.3 | 30 | **23.6** | **38.2** | **38.7** | 14.5 | 61.4 | **5.3** | **50.6** | **55.5** |
| PixFoundation† (8B)* (Ours) | ✗ | 52.0 | 52.0 | **52.0** | **56.1** | **54.0** | 62.2 | 72.2 | **54.3** | **64.4** | **68.1** |

Table 1: **PixMMVP** and **PixCV-Bench** benchmark evaluation of pixel-level MLLMs and baselines. We evaluate the VQA accuracy in the first and third probing (i.e., $\mathcal{A}$† and $\mathcal{A}$ resp.). Additionally, we evaluate pixel-level visual grounding with output segmentation in the first two probing (i.e., $\mathcal{M}$† and $\mathcal{M}$ resp.). *, **: models using Llama 3 (8B) and InternLM2 (7B) respectively, unlike the rest that are relying on Vicuna (7B and 13B) for the base LLM. - : indicates either the model can not be evaluated in that setting, or has low results below 1% showing complete failure in that setting. $\mathcal{S}$: denotes the score of the MLLM that is the harmonic mean of max($\mathcal{A}$, $\mathcal{A}$†) and. max($\mathcal{M}$, $\mathcal{M}$†). PixGr.: pixel-level grounding training. The *oracle* results are highlighted in red, the best and second best are bolded and underlined respectively.

the dorsal fin of the animal

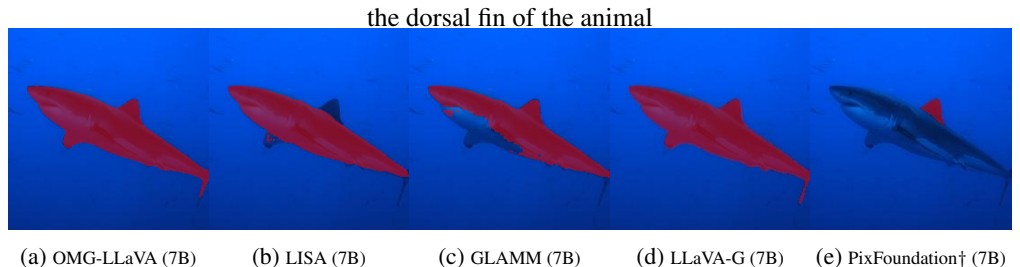

  (a) OMG-LLaVA (7B)     (b) LISA (7B)     (c) GLAMM (7B)     (d) LLaVA-G (7B)     (e) PixFoundation† (7B)

Figure 4: **PixMMVP** qualitative comparison in pixel-level visual grounding following the second probing technique. The referred expression is shown on top. It shows that mining for the grounding within the attention maps of vanilla MLLMs using their upper bound is better than MLLMs trained with pixel-level supervision, without degrading their VQA abilities. Thus, questioning whether the current training recipes and specialized decoders in pixel-level MLLMs are in the right direction.

the harmonic mean across the maximum of both pixel-level visual grounding and VQA,

$$\mathcal{S} = \frac{2}{\frac{1}{\max(\mathcal{A},\mathcal{A}\dagger)} + \frac{1}{\max(\mathcal{M},\mathcal{M}\dagger)}}. \qquad (1)$$

We mainly focus on evaluating four state-of-the-art pixel-level MLLMs; LISA Lai et al. (2024), GLAMM Rasheed et al. (2024), OMG-LLaVA Zhang et al. (2024b) and LLaVA-G Zhang et al. (2024a). For GLAMM we use two variants; the original model (GLAMM) and the one fine-tuned for region-level captioning (GLAMM-RegCap). For details on the models' weights, refer to App. A.

**Baselines and upper bound implementation details.** We evaluate: (i) the attend and segment (a+s), (ii) the *oracle* selection relying on the highest intersection over union in selecting the predicted masks (PixFoundation†), and (iii) the *automatic* selection (PixFoundation). These are implemented on top of three base MLLMs, which are LLaVA 1.5 (7B, 13B) Liu et al. (2024) and Cambrian-1(8B) Tong et al. (2024a). The automatic selection is implemented using GPT-4o. App. A has more details.

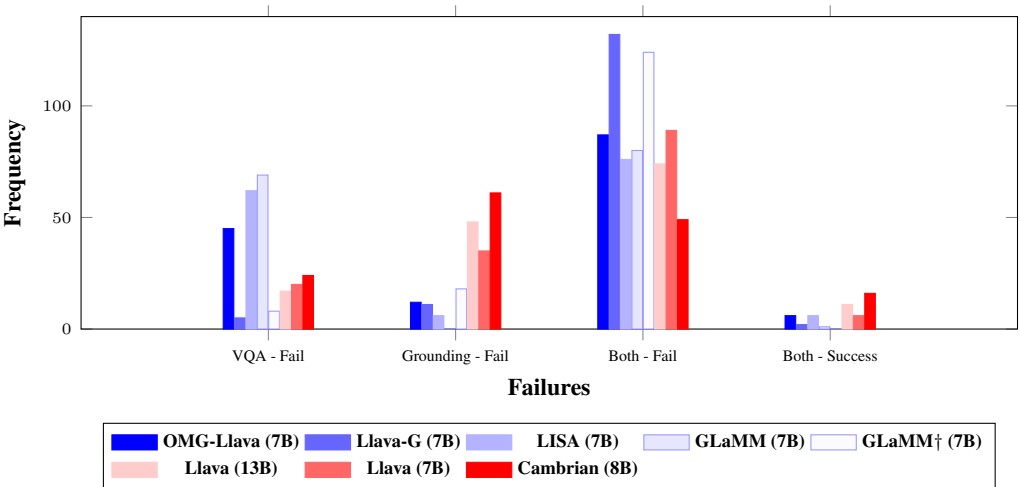

Figure 5: Frequency of failures in both visual grounding and VQA *vs.* VQA failures only *vs.* grounding only. For visual grounding, IoU $< 0.5$, is considered as a failure.

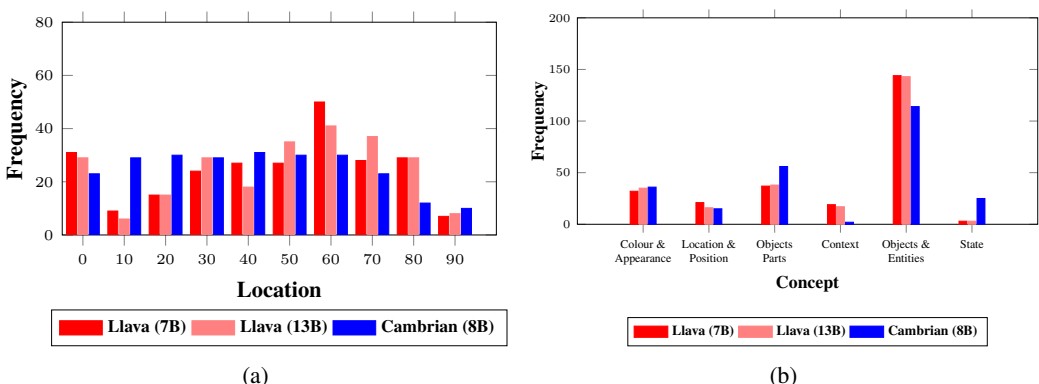

Figure 6: Analysis on when grounding emerges on PixMMVP benchmark using the three base MLLMs, LLaVA 1.5 (7, 13B) and Cambrian-1 (8B), that were not trained with pixel-level grounding supervision. We follow the second probing, then report the oracle selection. Analysis on: (a) the output location and (b) the output concept category, which coincides with the best segmentation.

## 4.2 Are the current pixel-level MLLMs heading in the right direction?

In order to answer this, we evaluate each of these pixel-level MLLMs capability in VQA in challenging tasks. Additionally, we evaluate their ability to visually ground the objects of interest in these questions. Table 1 shows the results on the challenging PixMMVP and PixCV-Bench. From the accuracy of VQA, MLLMs that are not trained with pixel-level grounding surpass their pixel-level counterpart with up to 14%. The best in pixel-level MLLMs score in this aspect is GLAMM-RegCap Zhang et al. (2024b) yet it has degraded ability to generate segmentation. On the other hand, when looking at pixel-level visual grounding, we find the best model, GLAMM Rasheed et al. (2024), has a weak ability in VQA or following instructions. Moreover, it shows LISA and LLaVA-G are mostly incapable of following the instruction to output the option letter reported in $\mathcal{A}$. OMG-LLaVA strikes the right balance in both VQA and pixel-level grounding with the highest score, $\mathcal{S}$, within pixel-level MLLMs. However, looking at the bottom three rows, the *oracle* confirms that MLLMs that were never trained with pixel-level grounding have the correct grounding within their learned attention maps, refer to Fig. 4. Additional qualitative analysis is in App. B. Looking at the final score, $\mathcal{S}$, the oracle variant, PixFoundation† (7B), outperforms the corresponding best pixel-level MLLM, OMG-LLaVA (7B), by a considerable margin, while the automatic outperforms it with up to 8% on PixMMVP. Furthermore, the attend and segment baseline Cao et al. (2024) lags behind our automatic method by more than 10%. Refer to App. C for additional results and App. D for failure analysis.

Finally, we evaluate whether the failures of these MLLMs occur in visual grounding, VQA or both. Figure 5 shows the frequency of failures per category, where the majority stem from failures in both, especially in the pixel-level MLLMs. The vanilla MLLMs perform better in the VQA than grounding.

**Summary.** In summary, pixel-level grounding supervision with specialized segmentation decoders degrades MLLMs ability in VQA and sometimes even their generalization in grounding. We show that MLLMs trained with pixel-level supervision lag behind vanilla MLLMs using simple mechanisms to extract grounding, and the *oracle* indicates there is an opportunity to improve this. Moreover, we show that grounding might not coincide with the noun phrase most similar to the referred expression, where our *oracle* upper bound and *automatic* baseline both surpass the attend and segment.

### 4.3 When does grounding emerge in MLLMs?

**When - location.** Taking into account the powerful performance of the *oracle* upper bound, it begs the question of when grounding emerges. We start by looking at when it emerges in terms of the location. We analyze the word/phrase location with respect to the full output text in terms of a percentage of its total length (i.e., 0% means the beginning of the text). Accordingly, Fig. 6a shows the location percentages histogram, binned at 10%, for the three base MLLMs reporting the oracle selection and evaluating on PixMMVP benchmark using the second probing. In the LLaVA 1.5 variants, the highest grounding is at the last 40%, while for Cambrian it is at the last 60%.

**When - concept.** For the second analysis, we look into the concept category that the correct output word/phrase corresponds to. The previous assumption in other works is that grounding emerges in the exact noun/noun phrase of the object of interest. Except our analysis confirms that this is not necessarily the case. We take the correct noun/noun phrase where the grounding emerges based on the *oracle* from all three variants, then we pass it to GPT-4o to request a grouping of these concepts. It result in six main groups, which are: (i) color and appearance, (ii) location and position, (iii) object parts, (iv) context and setting, (v) objects and entities, and (vi) State. We then prompt for each of the noun/noun phrases, GPT-4o, to categorize it within these six categories. The histogram of the occurrences of these concept categories is shown in Fig. 6b. It conveys that in certain scenarios, the correct output when grounding emerges can be describing the position or the color of the object, not necessarily the exact referring expression. Fig. 3 shows qualitative examples of these scenarios. We can see in PixMMVP up to 45% of the examples exhibit this phenomenon, referring to Fig. 6b and computing the percentage of examples that are not under the concept "Objects and Entities". Results for PixCV-Bench are provided in App. E with up to 27% of the examples showing similar behaviour.

**Random vs. best.** Our baselines rely on the maximum attention per output noun phrase to prompt SAM for the segmentation mask. Nonetheless, as a lower bound analysis, we evaluate the performance if we use a random point as a prompt instead. For fair comparison, we generate random points with the count of output masks that the oracle has to select among (i.e., the number of the output noun phrases). We conduct this ablation on PixMMVP using LLaVA 1.5 (7B) base MLLM, with random point prompts followed by the *oracle* selection among their SAM masks. Figure 3 prompts ablation, shows that random + oracle lags behind the correct one using the maximum point (i.e., First) with around 12%. More importantly, we confirm the stability of the results if we select the second-best or third maximum attention (i.e., Second and Third), which are on par with the maximum point.

**Summary.** In summary, we found that emergent grounding might not coincide with the input referring expression. We show that grounding in MLLMs can emerge in the noun phrase that corresponds to color, position or other characteristics of the object of interest.

## 5 Conclusion

We propose two benchmarks showing that pixel-level MLLMs degrade the ability in VQA and even grounding of fine-grained objects. Thus, our results question whether we are heading in the right direction with these models. Additionally, we provide powerful baselines with improved scores without training for pixel-level grounding. Our paired benchmarks and evaluation pave the road towards better interpretability and benchmarking efforts. We leave it for future work to investigate the use of pixel-level supervision, the training recipes and the use of specialized segmentation decoders when building pixel-level MLLMs, relying on our benchmarks.

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

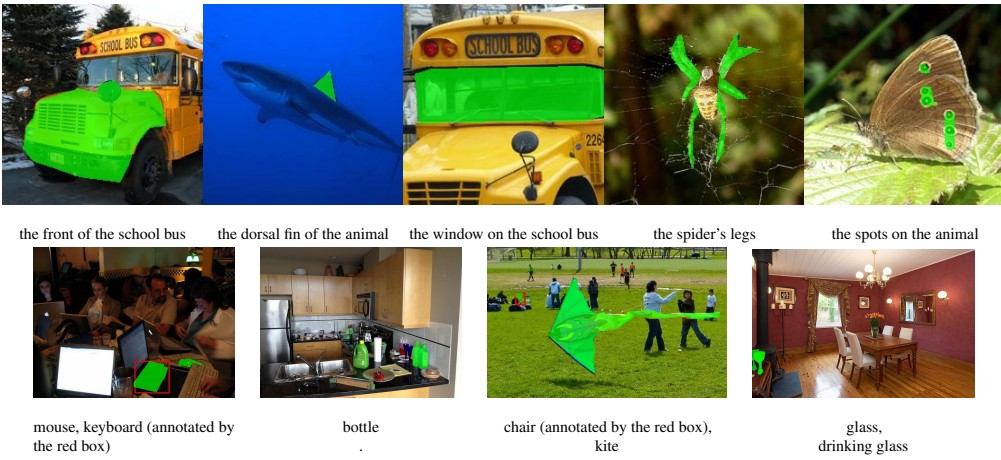

the front of the school bus    the dorsal fin of the animal    the window on the school bus    the spider's legs    the spots on the animal

mouse, keyboard (annotated by the red box)      bottle .      chair (annotated by the red box), kite      glass, drinking glass

Figure 7: Examples of ground-truth annotations for referring expressions in the respective object of interest in the question and their segmentation masks. First row: PixMMVP examples, Second row: PixCV-Bench examples. Ground-truth highlighted in green.

# A  Additional implementation details

In this section, we cover additional details about our proposed datasets and the implementation of the evaluation setup and baselines. We also refer to the output from the questions of the three probing techniques in the supplementary material for all the studied models.

**Datasets.** Our proposed datasets, PixMMVP and PixCV-Bench, are composed of ground-truth referring expressions describing the object of interest in the respective question and its segmentation mask. We show in Fig. 7 examples of these ground-truth annotations for both datasets. It shows the challenging scenarios in pixel-level visual grounding, which is strongly tied to the visual question answering task, since an integral part of answering these questions requires the grounding of the object/s of interest.

**Models.** We also detail the model checkpoints we use for the four pixel-level MLLMs and their variants, retrieved from HuggingFace Wolf et al. (2019) in Table 2. These also include the model checkpoints used for the base MLLMs that were not trained with pixel-level visual grounding. It is worth noting that for GLAMM we use two variants (FullScope and RegCap) since their base model (i.e., FullScope) has low performance in the visual question answering task. As such, we use the other variant for GLAMM that was fine-tuned for region-level captioning using RefCOCOg. Furthermore, we provide details on the *oracle* selection mechanism, we discard the cases where the ground-truth segmentation is all background in the when analysis, since there is no ground-truth grounding emerging to evaluate against. While in the quantitative and qualitative evaluation, we resort to simply not selecting any mask. These occur in a few cases in PixMMVP.

Additionally, we provide details on the SAM model that is used in the three baselines and upper bounds in our benchmarks, where we use the ViT-H variant. Finally, we provide an illustrative example of our automatic selection mechanism with the corresponding predictions on PixMMVP using LLaVA 1.5 (7B) in Fig. 8. Our automatic selection goes through an iterative process of prompting the selected MLLM, in our case GPT-4o, with N images highlighted with the predicted segmentation to select the best within each group of three. In the final stage, the best images are used to prompt the MLLM to select the final mask that best describes the object of interest. In the *oracle* upper bound, whenever the model is to be evaluated in a multiple object scenario, we take all the possible pairs of the masks and select the best pair based on the highest intersection over union.

**Evaluation.** We also provide the details on computing the visual question answering accuracy using GPT-4o in the first protocol Tong et al. (2024b). We use the following prompt: *"Given the following question <QUESTION>, the correct answer is <ANSWER>. Does the following answer correctly answers the question, answer: <RESPONSE>? Respond with a Yes/No"*. Note that all our inference and evaluation were conducted on an A600 84G GPU-equipped machine.

| Model Name | Model Checkpoint |
|---|---|
| LISA | xinlai/LISA-7B-v1-explanatory |
| GLAMM | MBZUAI/GLaMM-FullScope |
| GLAMM-RegCap | MBZUAI/GLaMM-RegCap-RefCOCOg |
| LLaVA-G | Haozhangcx/llava_grounding_gd_vp |
| LLaVA 1.5 (7B) | liuhaotian/llava-v1.5-7b |
| LLaVA 1.5 (13B) | liuhaotian/llava-v1.5-13b |
| Cambrian-1 (8B) | nyu-visionx/cambrian-8b |

Table 2: Hugging Face model checkpoints used in our benchmarks.

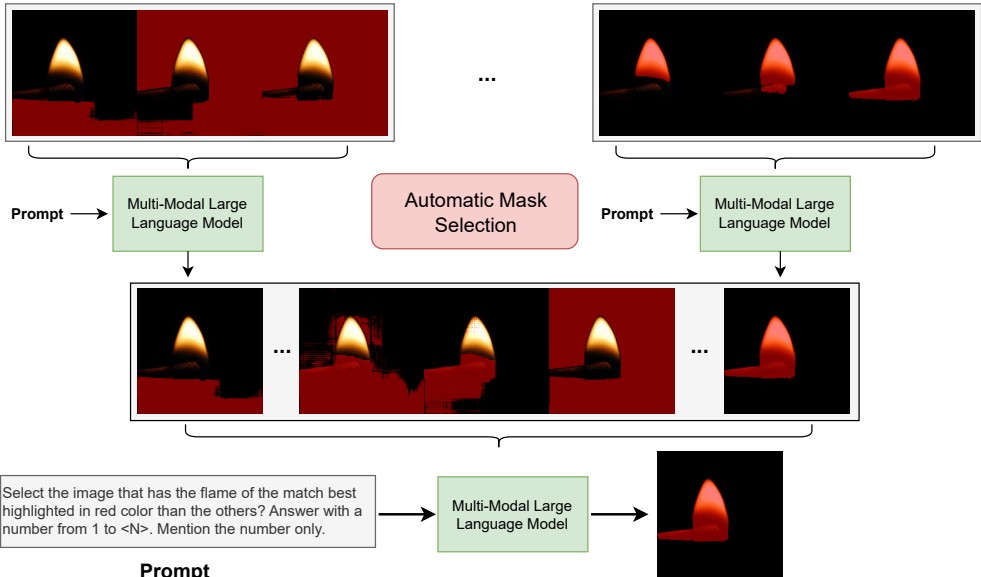

Figure 8: The automatic selection baseline, PixFoundation, which uses a simple mechanism of highlighting the predicted masks in red then prompting a multi-modal large language model to select the right mask from the group of highlighted images, followed by the final mask selection.

## B  Additional qualitative analysis

In this section, we provide a qualitative ablation of our baselines and a visualization of the attention maps that can show how vanilla MLLMs are reasoning on the question they are answering. Additionally, we provide qualitative examples showing when grounding emerges in these vanilla MLLMs. Finally, we provide more examples on PixMMVP and PixCV-Bench benchmarks.

### B.1  Baselines ablation

We show the qualitative ablation among the two baselines and upper bound using the best base MLLM Cambrian-1 (8B) in Fig. 9 on PixMMVP. The three confirm that there is grounding emerging in MLLMs that were not trained with pixel-level grounding supervision. Nonetheless, it shows that identifying when that grounding emerges is equally important in retrieving the best segmentation of the referring expression. The first baseline, attend and segment, assumes the alignment between the attention map that can be mined for the segmentation mask and the noun phrase that has the highest correspondence to the ground-truth category or noun phrase. Our findings quantitatively and qualitatively show otherwise, where grounding can emerge in different output tokens. It also shows the *oracle* upper bound for mask selection, PixFoundation†, exhibiting better segmentation than the attend and segment, confirming the aforementioned finding. Additionally, it shows that our simple automatic mechanism, PixFoundation, surpasses the attend and segment as well on PixMMVP.

Attend and Segment Cao et al. (2024) (Cambrian-1 (8B))

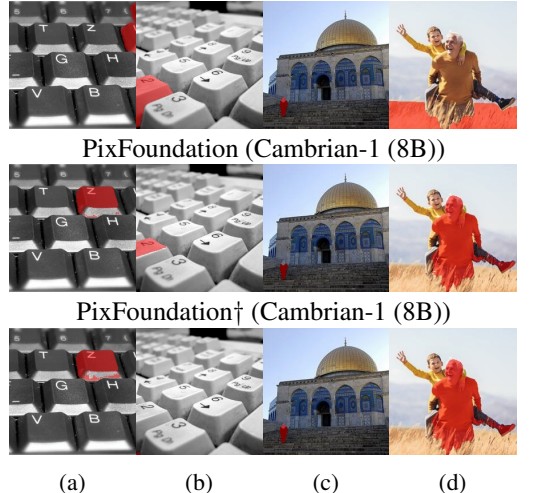

PixFoundation (Cambrian-1 (8B))

PixFoundation† (Cambrian-1 (8B))

(a)     (b)     (c)     (d)

Figure 9: Baselines and upper bound ablation using the base MLLM, Cambrian-1 (8B), ablating the different schemes for mask selection. We use the second probing to prompt the MLLM to identify the referred expression. The referring expressions for these examples are as follows: (a) the key "z", (b) the key "z", (c) people, (d) the elderly person. Predictions are highlighted in red.

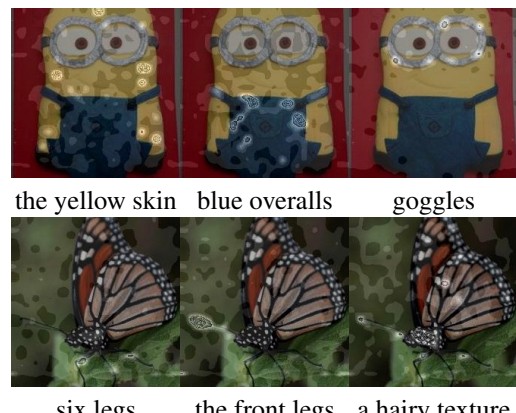

the yellow skin   blue overalls     goggles

six legs    the front legs  a hairy texture

Figure 10: Normalized attention maps visualization showing the noun phrase and its corresponding attention in the output text for two PixMMVP examples using Cambrian-1 (8B) base MLLM. While the attention maps can not be directly used as segmentation, yet it provides initial locations for the maximally attended pixels corresponding to what the model is looking at. In certain scenarios it exactly aligns with the noun phrase describing it as in the two examples. Yet in certain scenarios as we showed earlier, the grounding of the referred expression in question emerges with other noun phrases describing it.

## B.2 Attention maps visualization

In this section, we visualize the normalized attention maps, $\tilde{A}$, in Fig. 10. We show two examples for Cambrian-1 (8B) from PixMMVP using the first probing where we directly prompt the model with question and options. The first row shows outstanding ability to visually ground the different noun phrases from the output text. The full output text of the first row example is: *"The image provided is a cake designed to resemble a minion from the Despicable Me franchise. It is not a living creature and therefore cannot smile or have a tongue out. The cake is an inanimate object, crafted to mimic the appearance of a minion, which is a fictional character from the animated movie series. The design elements such as the **yellow skin, blue overalls, and goggles** are characteristic of the minions' appearance in the films."* The visualization shows how the maximally attended locations for the last three noun phrases correspond to the correct locations in the image.

The second output text corresponding to the example shown is; *"The butterfly's feet, also known as tarsi, are not distinctly visible in this image due to the angle and the butterfly's wings being open. However, we can infer their presence and approximate location. Monarch butterflies have **six legs**, with the hind legs being the longest and the **front legs** being the shortest. The legs are typically slender and have a **hairy texture**, which aids in gripping onto surfaces. In this image, the legs are likely located at the bottom of the butterfly's body, just below the abdomen, and are probably in contact with the leaf it is perched on."* The attention maps highlight what we suspect is a failure where the MLLM mistakes the antenna of the butterfly for front legs. Such hidden failures that do not necessarily affect the correctness of the answer, are still important to study and we believe our tool with the *oracle* upper bound can be used to inspect this further. Finally, we find that these attention maps in both examples are not sufficiently accurate to be used for segmentation directly, yet when paired with a powerful segmentation method like SAM it provides a good segmentation performance.

### B.3 When does grounding emerge?

We show additional examples of when grounding emerges in multi-modal large language models, specifically in the LLaVA 1.5 (7B) variant, using the second probing to prompt the model to segment what is in the referring expression. Figures 11, 12, 13 and 14 show the corresponding predicted masks for the grounding that emerged, highlighted in red with the maximum attention point as a black circle. Figure 3 shows the aforementioned four examples with the referred expression, the concept category and the noun phrase corresponding to the best grounding using the *oracle* selection and the full output text. It clearly shows that the correct output token can correspond to location or color, but not necessarily the ground-truth referring expression. While some of the noun phrases and their masks, from the SAM point prompting, correspond to what the noun phrase is describing. It is not always the case, for example, in Fig. 13 "the flame" was not able to highlight the correct object, yet it appeared in the noun phrase corresponding to the location "the top". While few scenarios might have the grounding coinciding with multiple noun phrases, such as in Fig. 11, "a butterfly" and "orange wings". Nonetheless, it is still an important insight that the segmentation can emerge corresponding to noun phrases that do not correspond to the exact referred expression. Our PixFoundation† serves as an interesting tool to interpret and understand how MLLMs work and reason to produce the final output with the *oracle* selection as an upper bound.

In summary, we provide four strong evidence that grounding can emerge corresponding to noun phrases that do not match the exact referred expression, as follows: (i) The attend and segment that rely on SpaCy embeddings lag behind our *automatic* and *oracle* mask selection, indicating that the noun phrases closest to the referred expressions are not necessarily where the optimal segmentation emerges. (ii) We show quantitative analysis on the location and the concept categories of the noun phrases where the grounding emerge that confirm the previous result. Where we show 45% of the examples in PixMMVP and 27% in PixCV-Bench have grounding emerging to noun phrases that are not describing objects and entities. (iii) We show qualitative analysis to confirm this further. (iv) We also provide the results for a simple analysis that compares the noun phrases text, where grounding is emerging, to the input referred expression text, where we find a mismatch between both with up to 92% in PixMMVP. However, the first two results better reflect the right metric to evaluate when grounding emerges, as they take into account noun phrases that might have similarities to the input referred expression with minor differences and same meaning.

### B.4 PixMMVP benchmark

Figure 15 shows additional results on PixMMVP benchmark comparing different pixel-level MLLMs with our *oracle* baseline using LLaVA 1.5 (7B). While GLAMM shows strong pixel-level visual grounding yet we have shown earlier that it is almost incapable of visual question answering which renders the model weak for general purpose tasks. On the other hand, OMG-LLaVA shows a better balance in pixel-level visual grounding and visual question answering as previously detailed. Nonetheless, the simple mining of attention maps from LLaVA 1.5 (7B) using the *oracle* selection which we call PixFoundation† shows the strongest capability in both grounding and VQA. In fact, certain MLLMs that were trained with pixel-level visual grounding, such as LISA, have degraded the performance with respect to the hidden information already existing in powerful MLLMs that were not trained with such supervision.

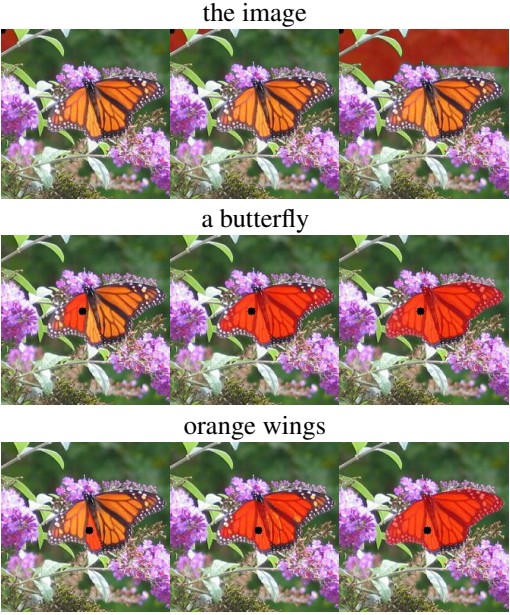

Figure 11: First example of when grounding emerges, corresponding to Image 1 in Fig. 3. Each row has the corresponding noun phrase on top and three potential SAM predicted masks highlighted in red using the maximum attention point of this noun phrase as a point prompt, highlighted as a black circle. It shows the output from mining the attention maps for pixel-level grounding using LLaVA 1.5 (7B) base MLLM.

| Model Name | Probing | Output Length | # Noun Phrases |
|---|---|---|---|
| LLaVA 1.5 (7B) | First | 44.2 | 2.3 |
| LLaVA 1.5 (13B) | First | 45.3 | 2.4 |
| Cambrian-1 (8B) | First | 313.8 | 15.2 |
| LLaVA 1.5 (7B) | Second | 92.6 | 5.2 |
| LLaVA 1.5 (13B) | Second | 97.2 | 5.5 |
| Cambrian-1 (8B) | Second | 561.3 | 27.3 |

Table 3: The average output length across PixMMVP dataset for the three base MLLMs using the first and second probing techniques.

### B.5 PixCV-Bench benchmark

Figure 16 shows qualitative results on PixCV-Bench. It shows that pixel-level MLLMs struggle with segmenting the object annotated by the red box unlike our *oracle* baseline, PixFoundation†. Indeed the attention maps from these MLLMs are looking at the right object annotated by the red box without receiving any pixel-level grounding supervision during training.

## C   Analysis on the output length

In this section, we provide additional analysis on the output length on average through PixMMVP dataset using the first and second probing schemes. Specifically, we report the output length as the number of characters in the output, and the number of noun phrases extracted from it. The reason to study this, since it has relation to the number of noun phrases and consequently the number of masks our baselines are selecting among. Table 3 shows the average output length computed across PixMMVP dataset, comparing the three base MLLMs. We notice that Cambrian-1 (8B) generates longer outputs with a considerable margin than LLaVA variants. Hence, we believe the superiority of the *oracle* upper bound with Cambrian-1 in the grounding has strong correlation to producing longer outputs with more attention maps to mine and select from, than LLaVA variants. Nonetheless, it makes it more challenging for the automatic baseline.

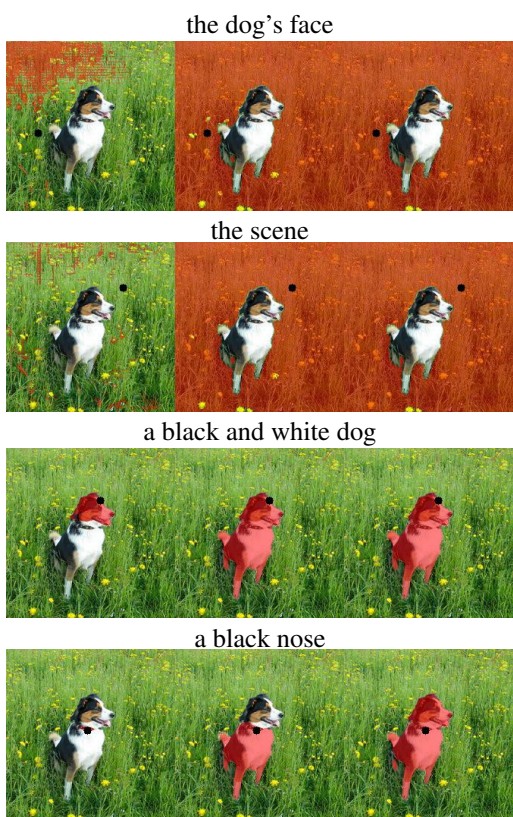

Figure 12: Third example of when grounding emerges, corresponding to Image 6 in Fig. 3. Each row has the corresponding noun phrase on top and three potential SAM predicted masks highlighted in red using the maximum attention point of this noun phrase as a point prompt, highlighted as a black circle. It shows the output from mining the attention maps for pixel-level grounding using LLaVA 1.5 (7B) base MLLM.

## D    Failure Cases Analysis

In this section, we conduct additional failure case analysis of pixel-level MLLMs and our baselines qualitatively and quantitatively.

### D.1    Failures in Visual Question Answering

We start with a fine-grained quantitative analysis of how the studied models perform across PixMMVP and PixCV-Bench. For PixMMVP we follow their scheme to identify the nine visual patterns and report the model's accuracy with each pattern in Fig. 17. Similarly, we show fine-grained analysis relying on the tasks for the two datasets (ADE20K and COCO) in Fig. 18.

PixMMVP results show that the majority of pixel-level MLLMs, highlighted in blue, suffer in the state, orientation and quantity related tasks. On the other hand, relational context, color and presence of features show the best performance with pixel-level MLLMs. Nonetheless, across all the visual patterns, the MLLMs that were not trained with pixel-level supervision persistently surpass these pixel-level MLLMs with a considerable margin. PixCV-Bench, similarly shows the count task is more challenging than the relational positioning. It also shows that ADE20K dataset serves as a more challenging dataset than COCO.

### D.2    Failures in Pixel-level Visual Grounding

Finally, we show qualitatively the failure cases of the *oracle* upper bound in Fig. 19. It shows failures in segmenting all the object instances in the first row, since the current point prompting assumes one

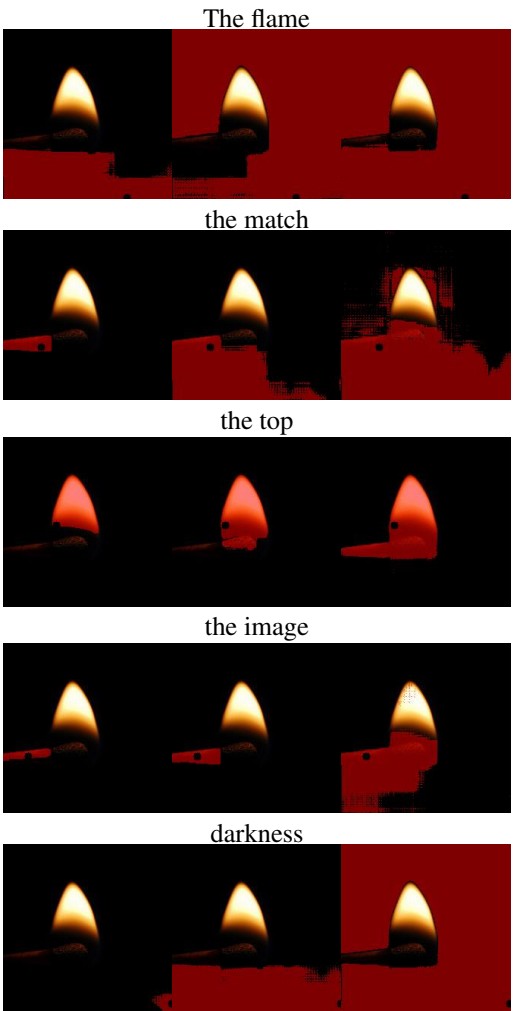

Figure 13: Second example of when grounding emerges, corresponding to Image 3 in Fig. 3. Each row has the corresponding noun phrase on top and three potential SAM predicted masks highlighted in red using the maximum attention point of this noun phrase as a point prompt, highlighted as a black circle. It shows the output from mining the attention maps for pixel-level grounding using LLaVA 1.5 (7B) base MLLM.

connected component corresponding to each expression. However, certain scenarios, such as the image with the spots on the animal, can lead to these failures in the oracle even when the localisation of some of these is correct. Mechanisms that solve this multi instance scenarios of the same object are left for future work.

Another failure occurring such as in the second row stems from ambiguity in the referring expression itself or failures from SAM identifying the separation between the wall and the ceiling. Hence, the oracle upper bound is generally inheriting SAM failures. However, its main purpose of showing that the hidden information within powerful MLLMs is sufficient to perform pixel-level grounding is achieved, and even surpassing pixel-level MLLMs without degrading their VQA abilities.

## E    Additional quantitative analysis

### E.1    Automatic baseline using open-source models

In our *automatic* baseline, we replace GPT-4o, which is a closed source model, with another open-source model, in our case Cambrian-1 (8B). Table 4 shows the results on PixMMVP for PixFoundation

The minute hand

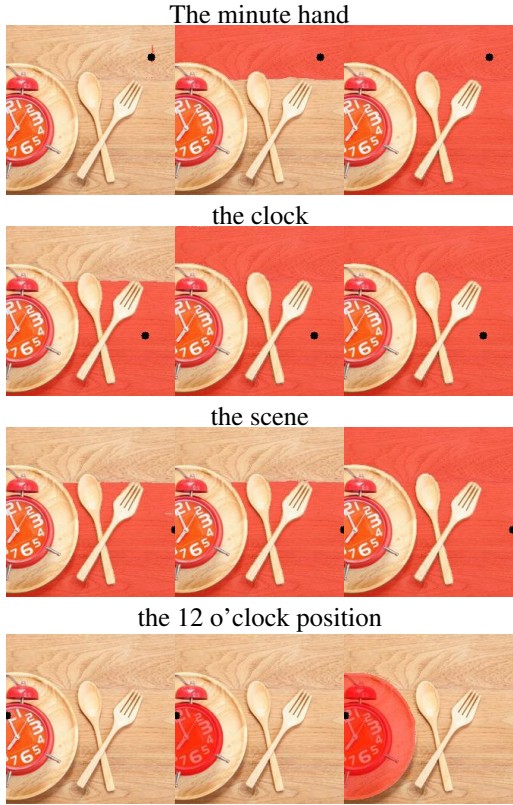

the clock

the scene

the 12 o'clock position

Figure 14: Fourth example of when grounding emerges, corresponding to Image 161 in Fig. 3. Each row has the corresponding noun phrase on top and three potential SAM predicted masks highlighted in red using the maximum attention point of this noun phrase as a point prompt, highlighted as a black circle. It shows the output from mining the attention maps for pixel-level grounding using LLaVA 1.5 (7B) base MLLM.

| Method | PixMMVP | | | | |
|---|---|---|---|---|---|
| | $\mathcal{A}$† | $\mathcal{A}$ | $\mathcal{M}$† | $\mathcal{M}$ | $\mathcal{S}$ |
| OMG LLaVA (7B)** | 12.0 | 12.0 | 17.8 | 38.0 | 18.2 |
| LLaVA 1.5 (7B) + (a+s) | 27.3 | 28.0 | 11.1 | 11.2 | 16.0 |
| LLaVA 1.5 (13B) + (a+s) | 39.3 | 30 | 9.8 | 11.4 | 17.7 |
| Cambrian (8B)* + (a+s) | 52.0 | 52.0 | 14.3 | 15.1 | 23.4 |
| PixFoundation⋆ (8B)* (Ours) | 52.0 | 52.0 | 17.2 | 18.9 | **27.7** |

Table 4: **PixMMVP** comparison of pixel-level MLLMs to our automatic baseline that relies on Cambrian-1 (8B), an open-source model, for the automatic selection (PixFoundation⋆). Instead of using GPT-4o, which is closed source. Best results are bolded.

*automatic* baseline that still surpasses the best pixel-level MLLM, OMG-LLaVA, without the use of pixel-level supervision. More importantly, this baseline confirms that even with the use of a self-contained model as Camrbian-1, without additional help from GPT-4o in a training-free mechanism, it can still compete with these pixel-level supervised models.

### E.2 When grounding emerges - PixCV-Bench

In Fig. 20a we show the analysis on when grounding emerges on PixCV-Bench in terms of the frequency of the grounding location. It is worth noting that PixMMVP is more challenging than PixCV-Bench, evidently from the reported IoU and accuracy metrics on both with respect to Table 1. It seems on the less challenging dataset PixCV-Bench, grounding tends to emerge frequently near the beginning of the output. This might relate to PixMMVP being more challenging in terms of the level

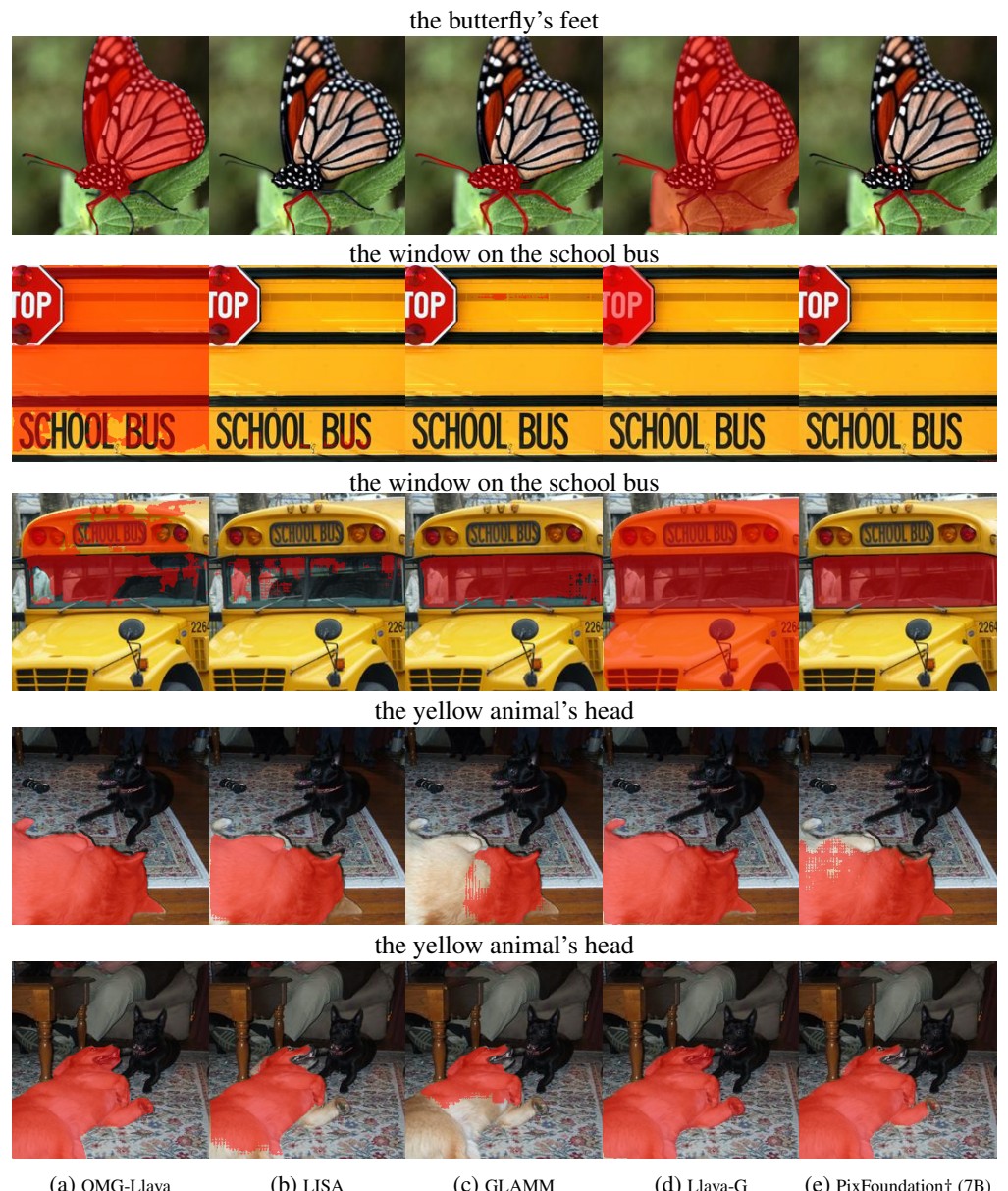

the butterfly's feet

the window on the school bus

the window on the school bus

the yellow animal's head

the yellow animal's head

|   |   |   |   |   |
|---|---|---|---|---|
| (a) OMG-Llava | (b) LISA | (c) GLAMM | (d) Llava-G | (e) PixFoundation† (7B) |

Figure 15: **PixMMVP** qualitative comparison between the pixel-level visual grounding following the second probing. The referred expression used in the segmentation is shown on top of each row. It shows persistently that mining for the grounding within attention maps of MLLMs that were not trained with pixel-level grounding supervision and using the oracle selection outperforms the pixel-level MLLMs. It clearly shows the oracle excels in identifying fine-grained object parts and descriptions that other pixel-level MLLMs are not necessarily capable of. The second best performance is GLAMM, yet we showed it is completely incapable of performing visual question answering unless fine-tuned for the region captioning task at which then it loses its grounding ability.

of reasoning than PixCV-Bench or the fact that PixMMVP poses a harder referring segmentation task than PixCV-Bench, which is mostly using the class names. Another difference is that PixMMVP is out of the distribution of the seen datasets for these MLLMs. However, the consistent finding among both datasets is that grounding can emerge coinciding with various concept categories, whether location, color or state, as shown in Fig. 20b. It shows that up to 27% of the examples in PixCV-Bench exhibit this behaviour. Note that across this analysis, we compute the frequency per object in the referred expression corresponding to the visual question. Hence, if we have two objects in one visual question,

cell phone

mouse and keyboard (annotated by the red box)

sports ball and person (annotated by the red box)

chair and cell phone

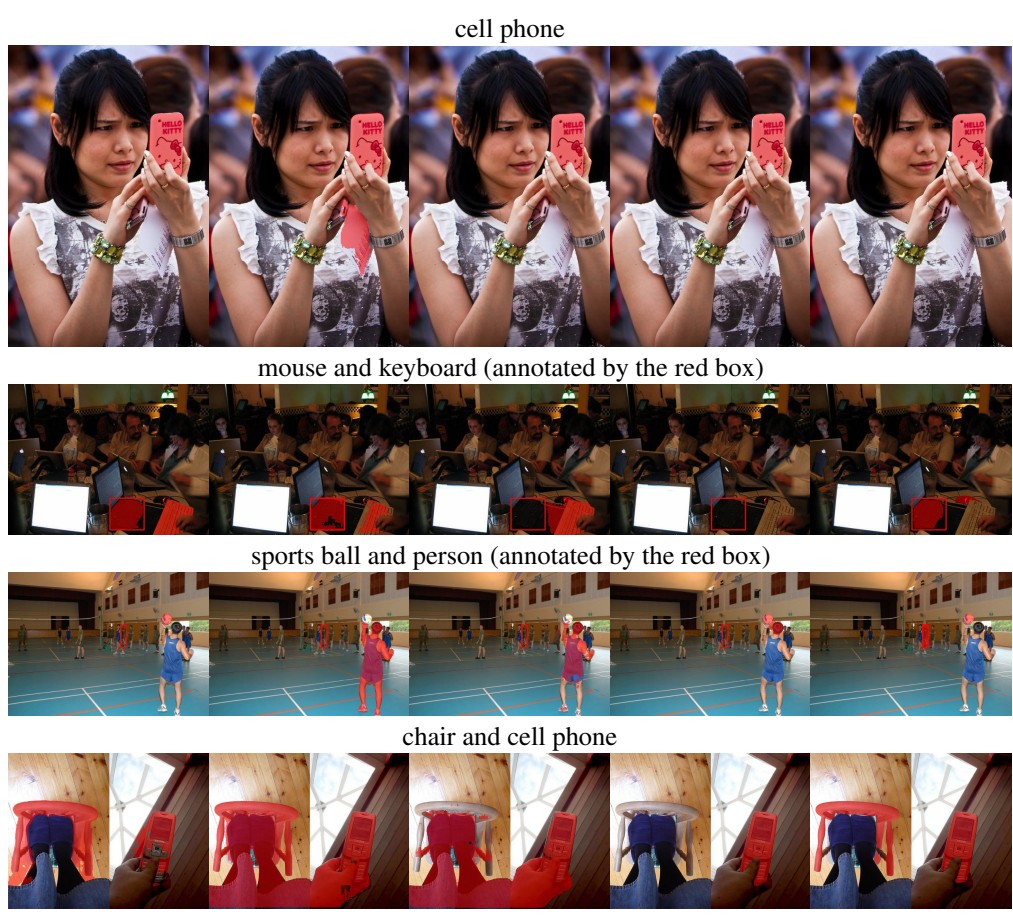

(a) OMG-Llava     (b) LISA     (c) GLAMM     (d) Llava-G     (e) PixFoundation† (7B)

Figure 16: **PixCV-Bench** qualitative comparison between the pixel-level visual grounding following the second probing. The referred expression used in the segmentation is shown on top of each row. It shows similar to PixMMVP that mining for the grounding within MLLMs that were not trained with pixel-level grounding supervision paired with the oracle selection outperforms pixel-level MLLMs.

such as in the relative positioning questions, each object's concept, corresponding to the emergence, is computed as part of our analysis.

## F   Licences and Assets

We use the MMVP and CV-Bench (2D) that were provided in their original works Tong et al. (2024b,a). The first is licensed under a MIT License that allows its use without restriction for research purposes. The second refers to the OpenAI Terms of Use for the instruction tuning dataset, which we do not employ and the specific licenses for base language models for checkpoints trained using the dataset (e.g. Llama community license for LLaMA-3, and Vicuna-1.5). They do not impose any additional constraints beyond those stipulated in the original licenses. Finally, all the studied models' trained weights were retrieved from HuggingFace as detailed earlier.

## G   Impact Statement

Multi-modal large language models are widely used in various applications, such as robotics, medical image processing and remote sensing. The pixel-level understanding within such MLLMs is necessary for such applications that require the localization and even in certain scenarios the delineation of the boundaries for the objects of interest. It is even more important to maintain a good chat performance

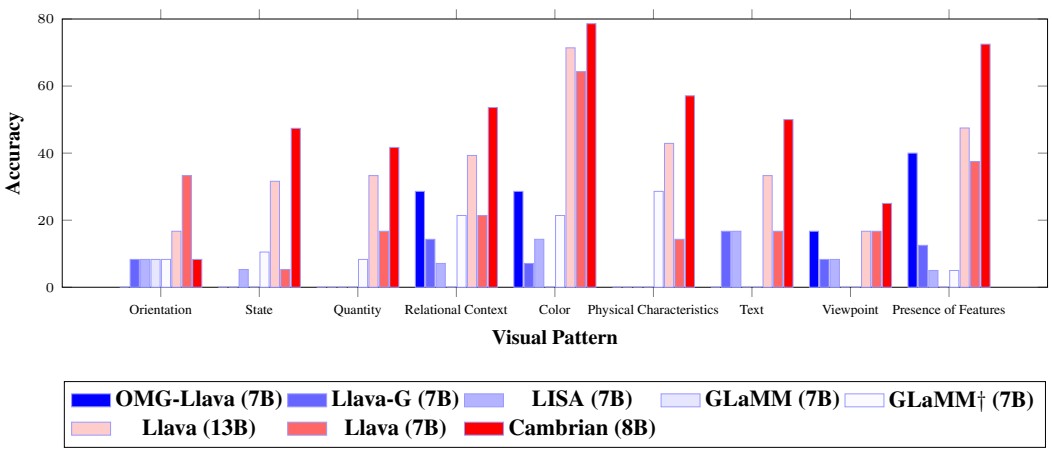

Figure 17: Fine-grained analysis of the studied models performance across the different visual pattern in PixMMVP showing the model's accuracy with each pattern.

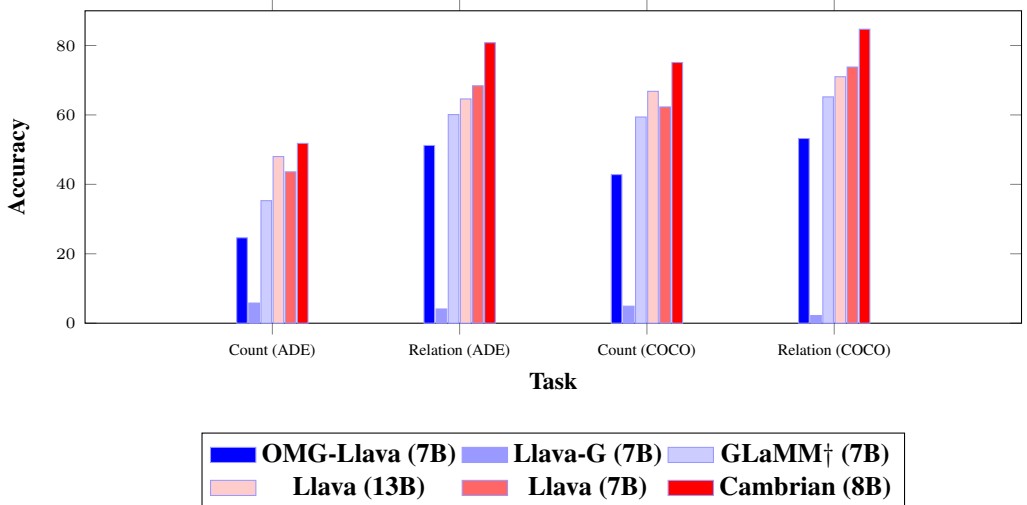

Figure 18: Fine-grained analysis of the studied models performance across the different visual patterns in PixCV-Bench (ADE20K and COCO), showing the model's accuracy with each pattern.

and visual question answering ability in such applications as well. In our work, we have investigated the shortcomings of pixel-level MLLMs while providing more challenging benchmarks for these, to improve them further.

However, as with many other AI advancements there are risks that could be entailed from the deployment of such models. There could be inherent biases emerging in such pixel-level MLLMs impacting various under-represented groups. We think that our benchmarking efforts and providing a tool to understand the pitfalls in the understanding and reasoning of these models could be an initial direction for mitigating such biases. Nonetheless, we leave it for future work to explore this further.

## H   Limitations

Note that our training-free baselines do entail a computational overhead with the use of the mask selection process. Nonetheless, the benefit from exploring what is already learned in these MLLMs through mining the attention maps with an understanding of when grounding emerges, provides greater benefit to interpretability. Where we believe interpretability of MLLMs is a crucial aspect when following a responsible approach to AI. Additionally, these baselines are mainly designed as

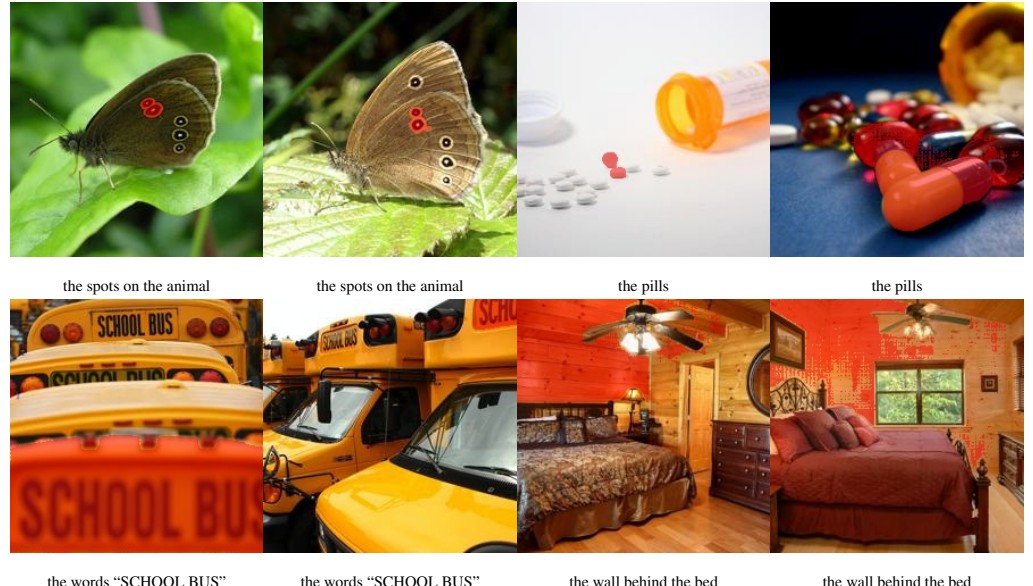

| the spots on the animal | the spots on the animal | the pills | the pills |

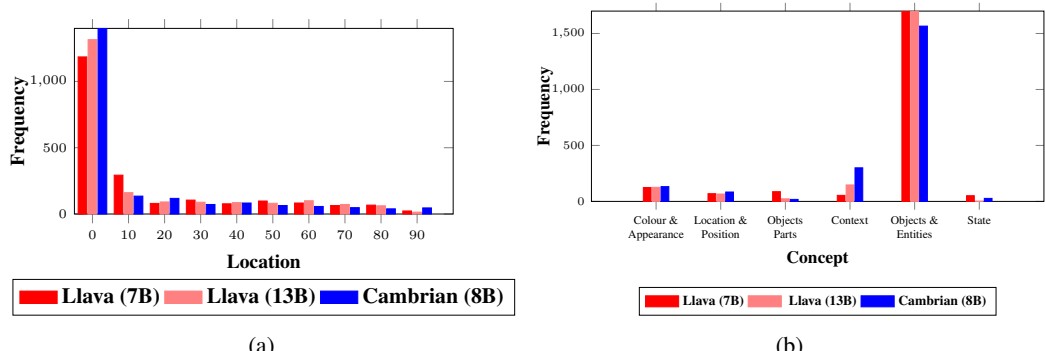

| the words "SCHOOL BUS" | the words "SCHOOL BUS" | the wall behind the bed | the wall behind the bed |

Figure 19: Failures of the *oracle* upper bound, PixFoundation†, using Cambrian-1 (8B) as base MLLM on PixMMVP. It shows the failures mostly emerge in quantity or counting tasks. It also shows that the upper bound is inheriting SAM failures and the ambiguity arising in the referred expression itself, e.g., "the wall behind the bed", which direction does "behind" indicate.

(a)    (b)

Figure 20: Analysis on when grounding emerges on PixCV-Bench benchmark using the three base MLLMs, LLaVA 1.5 (7, 13B) and Cambrian-1 (8B), that were not trained with pixel-level grounding supervision. We follow the second probing then report the oracle selection. Analysis on: (a) the output location and (b) the output concept category, that coincides with the best segmentation.

599  strong baselines in our paired benchmarks and to showcase the shortcomings in the current pixel-level
600  MLLMs.

