# OpenReview forum: "PixFoundation: Are We Heading in the Right Direction with Pixel-level Vision Foundation Models?"
_NeurIPS.cc/2025/Conference — Submitted to NeurIPS 2025_

### Official Review · Reviewer_E95z · 2025-06-29

**Clarity:** 2
**Significance:** 2
**Originality:** 2
**Rating:** 2
**Confidence:** 4

**Summary:**

This paper investigates the current capabilities of MLLMs on VQA tasks, with a particular focus on their ability to provide visually grounded answers. To facilitate this analysis, the authors construct two new benchmarks by extending existing datasets, MMVP and CV-Bench, originally introduced by Tong et al. (Eyes Wide Shut, CVPR 2024; Cambrian-1, NeurIPS 2024). The new benchmarks include additional segmentation masks that indicate the pixel regions necessary to answer each question, enabling evaluation of grounding behavior.


Based on the benchmarking results, the paper challenges the existing pixel-level MLLMs and makes several key claims:
- Pixel-level supervision in MLLMs degrade VQA performance.
- MLLMs trained without pixel-level grounding supervision can outperform state-of-the-art methods on VQA and grounding.
- Vanilla MLLMs - without segmentation decoders or pixel-level grounding supervision - can achieve superior performance when evaluated using a harmonic mean of VQA accuracy and grounding quality.
- The authors introduce a lightweight method for extracting segmentation masks from vanilla MLLMs. Unlike recent work such as Cao et al. (Emerging Pixel Grounding in Large Multimodal Models Without Grounding Supervision, ICLR 2025), their approach relies on identifying token attention corresponding to descriptive cues (e.g., appearance or location) rather than object nouns.

**Questions:**

Several clarification questions to ensure my understandings.

- Clarify “some of these methods” (Line 7). In multiple instances, the paper uses vague phrases such as “some of these” without specifying which models or approaches are being referenced. This ambiguity makes me have to guess what you are referring to. Please explicitly list or define the methods to help the readers follow your arguments and comparisons more clearly.

- Elaborate on “reasons for failures” (Line 12). Could you provide concrete examples or categories of failure modes? Also, are there any insights or hypotheses on how future models might overcome these failures?

- Line 18: "We show that grounding can coincide with object parts, its location, appearance, context or state, where we show 27-45% of the examples in both benchmarks exhibit this phenomenon." What characterizes the remaining >50% of examples? Do they mostly correspond to object nouns, or are there other patterns?

- Line 66 - Line 68: "Our mechanism uses the observation that grounding can emerge corresponding to different output tokens describing the object’s appearance or location, not necessarily the exact text of the object of interest, which we call PixFoundation." Is this grounding influenced by SAM’s inherent preferences for certain types of point prompts? Could it be that the descriptive tokens happen to produce better SAM prompts by coincidence rather than model grounding ability?

- Interpretability of token-location connection (Lines 257–263): Figure 6(a) analyzes the distribution of grounding across the output token positions in free-form MLLM responses. To me, it looks like a relatively uniform distribution, what actionable insight is being drawn from this analysis? Do you intend to use this distribution to inform the mask selection strategy?

**Ethical Concerns:**

["NO or VERY MINOR ethics concerns only"]

**Final Justification:**

I encourage the authors to consider the feedback provided by the reviewers and to work on improving the way of how to present your ideas and observations. Some of the experimental results are interesting, while the paper does not yet feel fully ready for publication. With further refinement, it could make a more compelling work.

**Limitations:**

yes

**Paper Formatting Concerns:**

no concerns.

**Quality:**

2

**Strengths And Weaknesses:**

**Strengths**
- This work extends two existing benchmarks by providing additional segmentation masks to indicate the pixel regions necessary to answer questions. While GPT was used to accelerate the annotation process, human verification was still involved to ensure data quality. I saw a large supplementary zip file containing the annotated data. Although I haven’t walked through all examples in detail, the additional annotations appear to be a valuable resource for the community to evaluate and better understand MLLMs’ grounding behavior.

**Weaknesses**

Overall, I do not think this work is ready for readers, primarily due to the rooms to increase research depth of methodology, provide rigorous evaluation, and especially provide actionable insight (if the claim is that existing pixel-level foundation models are not heading the right direction, what are your suggestions, how should we do the MLLM design and training with pixel-level supervision? The paper is too ambiguous on actionable items).

- The paper raises the question, “Are we heading in the right direction?”, but does not offer a concrete alternative or provide actionable next steps on what direction should be taken. If the claim is that current methods are not heading correct direction, the paper should articulate the gaps more clearly and propose steps to address them. The proposed solution is to extract attention maps from vanilla MLLMs and using SAM (which itself is large-scale trained with pixel-level supervision) and GPT-4o for post-hoc segmentation and mask selection. The proposed solution does not present a principled technical alternative. As such, it falls short of addressing the directional question posed by the authors.

- The benchmarking methodology does not look rigorous. Models are broadly grouped into two categories: (1) MLLMs with pixel-level supervision, and (2) vanilla MLLMs without it. However, models vary significantly in terms of training data, scale, intended use cases, and favorite prompts. For example, LISA is designed for segmenting specific objects based on user input (e.g., “please segment…”), while GLaMMa focuses on captioning with interleaved segmentation (e.g., “Describe the image in detail…”). These models were not instruction fine-tuned to handle multiple-choice (a)(b)(c) or yes/no VQA formats. Such discrepancies can heavily influence accuracy, making it problematic to draw strong conclusions about the value of pixel-level supervision. Hence, the claims in this paper make me feel unjustified given the number of confounding factors.

- Building on the above, the evaluation protocol and metric in Eq. (1) lack rigor. GPT-4o is used as part of the evaluation process, yet no effort is made to isolate or control for its influence. Factors like model versioning, conversation history, and response randomness can all affect the outcome. A standalone, well-defined evaluation metric is essential for reproducibility and for enabling future work to build on these results.

- The claim that “MLLMs not trained with pixel-level grounding or segmentation decoders can outperform others” (from Line 42) is not clearly supported. From my reading, the strong performance reported relies on assembling multiple foundation models. The final grounding masks are generated by SAM (which is trained with pixel-level supervision), and GPT-4o is then used to select the most relevant one. In essence, this suggests that vanilla MLLMs struggle with precise grounding. They may attend to multiple plausible regions but cannot reliably identify the correct one and output it. Therefore, another foundation model GPT-4o comes in for the mask selection.

- Overall, the paper looks to me is incomplete. While it introduces useful benchmarks and makes some interesting observations, the analysis remains at a surface level, and no solid solution is proposed to address the gaps identified for pixel-level MLLMs model design and training strategies. This feels more like a preliminary investigation than a fully developed research contribution.

---

> ### Author Rebuttal · Authors · 2025-07-25
>
> Response provided under Reviewer 1 comment.

---

### Official Review · Reviewer_1dPy · 2025-07-01

**Clarity:** 2
**Significance:** 2
**Originality:** 3
**Rating:** 2
**Confidence:** 4

**Summary:**

This paper benchmarks pixel-level MLLMs' VQA and grounding ability. Instead of evaluating on standard grounding benchmarks or VQA benchmarks, this paper proposed PixMMVP and PixCV-Bench for comprehensive VQA and grounding evaluation. Evaluating pixel-level supervised and vanilla MLLMs on these benchmarks, they find that pixel-supervised MLLMs may not perform as well as vanilla MLLMs in grounding. They further propose a plugin module, PixFoundation, which extracts segmentation maps from MLLMs. Experiments and analyses are conducted.

**Questions:**

- What is the pixel-level grounding supervision? How are the MLLMs trained?
- Why is the pixel-level grounding so important? Why does the paper also study referring expression comprehension? Why is answering questions with segmentation the right indicator of grounding and VQA abilities, compared to separate standard benchmarks?
- Can a method similar to PixFoundation help the pixel-level MLLMs in the proposed new task? Or is there a way to fine-tune or off-the-shelf test both pixel-level and vanilla MLLMs?
- Could you provide the VQA and grounding (both REC and RES) performance of the selected models on standard benchmarks?

**Ethical Concerns:**

["NO or VERY MINOR ethics concerns only"]

**Final Justification:**

This work addresses an interesting task; however, the scope may be overstated or insufficiently defined. More nuanced clarification and additional experiments are recommended.

**Limitations:**

The paper tries to propose an interesting new task. However, they overstate the scope and fail to explain concepts clearly. The motivation possibly has fundamental flaws, and the comparisons seem unfair.

**Quality:**

2

**Strengths And Weaknesses:**

Strengths
- The paper points out an interesting task that combines grounding and VQA.
- They curate VQA and Grounding paired benchmarks, PixMMVP and PixCV-Bench, which provide referring segmentation given a question.
- They use the proposed benchmarks to evaluate pixel-level and vanilla MLLMs, revealing some unintuitive findings, such as the pixel-level supervised MLLMs may not outperform the vanilla variants in grounding tasks.
- They propose PixFoundation, a way to extract segmentation from MLLMs.

Weaknesses
- The writing could be improved. Some parts are misleading and confusing, and some necessary definitions are not well-explained. For example, the title "Are we heading in the Right Direction with **Pixel-level Vision Foundation Models**" sounds like they work on Vision Foundation models like ResNet or ViT, but they only study MLLMs. Moreover, the pixel-level grounding supervision is not a common term in the grounding community. The task looks relevant to referring expression segmentation. Preliminaries and definitions are essential to help the readers understand the background.
- The paper aims to investigate whether "the pixel-level MLLMs trained with full grounding supervision are heading in the right direction to improve both grounding and VQA" (ln 38-39). However, this motivation may suffer from logical flaws.

a) The importance of pixel-level MLLMs is questionable. The state-of-the-art or dominant models do not seem to be trained with pixel-level supervision. Those general MLLMs have superior performance on various tasks, including VQA and grounding.

b) The second part is the correlation between the "right direction" and performing "grounding and VQA" at the same time. Not to mention, the grounding ability studied in this paper refers to expression segmentation only. Combining grounding and VQA into a new task can be interesting, but it does not indicate that evaluating grounding and VQA ability through standard benchmarks separately is convincing.

- A task-specific module helps well-established, powerful MLLMs outperform pixel-level MLLMs, but this is not completely fair. The paper proposes a new task that has not been seen during the training of MLLMs. Therefore, fine-tuning or a task-specific module is intuitively helpful in this case. Vanilla MLLMs are well-trained and are well-known for superior performance on various tasks. It is not surprising that they outperform pixel-level MLLMs after integrating a task-specific module. The unfair setting makes the findings doubtful.

---

> ### Author Rebuttal · Authors · 2025-07-25
>
> Response provided under Reviewer 1 comment.

---

> > ### Comment · Reviewer_1dPy · 2025-08-07
> >
> > I thank the authors for their effort.
> >
> > After reading through other reviews and discussions, the doubts on the "wrong"/"right" directions are unanimous. The rebuttal fails to provide more standard VQA/grounding comparisons or convince the necessity of this benchmark with the existence of popular VQA, RES/REC benchmarks.
> >
> > That being said, this work shows promise, but it is not ready at the current stage.

---

### Official Review · Reviewer_1PJW · 2025-07-03

**Clarity:** 1
**Significance:** 2
**Originality:** 2
**Rating:** 2
**Confidence:** 4

**Summary:**

This paper addresses an important and timely question about the effectiveness and direction of pixel-level grounding in multi-modal large language models (MLLMs). The authors present two novel benchmarks, PixMMVP and PixCV-Bench, specifically designed to evaluate the paired visual question answering (VQA) and pixel-level grounding capabilities of MLLMs. Additionally, they introduce a baseline method, PixFoundation, aiming to analyze the emergent grounding capabilities of MLLMs that have not been explicitly trained with pixel-level supervision.

**Questions:**

Please address the weaknesses above.

**Ethical Concerns:**

["NO or VERY MINOR ethics concerns only"]

**Final Justification:**

The paper introduces valuable new benchmarks for evaluating pixel-level grounding in MLLMs, but the core claims about the “wrong direction” of pixel-level models are not sufficiently supported by controlled experiments or causal analysis. The evaluation lacks rigorous apple-to-apple comparisons with strong existing baselines, making it difficult to attribute observed performance differences to inherent limitations rather than implementation choices. Without stronger empirical evidence and broader dataset validation, the conclusions remain speculative, and the work does not yet meet the threshold for acceptance.

**Limitations:**

yes

**Quality:**

2

**Strengths And Weaknesses:**

Strenghts:
Two new paired benchmarks (PixMMVP and PixCV-Bench) are introduced, providing valuable resources for future research.

Weaknesses:
It is unclear what the authors mean by MLLMs not heading in the right direction. Is it because pixel-level MLLMs have shown degradation in their capabilities and chat performance, or because MLLMs without pixel-level grounding supervision can outperform the state of the art in grounding tasks? If the authors mean the former, degradation in VQA performance might be acceptable due to multi-task joint training conflicts or limited model capacity rather than an indication of moving in the wrong direction. If the authors mean the latter, the new proposed method should be rigorously compared against existing widely-used grounding datasets to validate its superiority.

---

> ### Author Rebuttal · Authors · 2025-07-25
>
> Response provided under Reviewer 1 comment.

---

### Official Review · Reviewer_xoMW · 2025-07-05

**Clarity:** 3
**Significance:** 2
**Originality:** 3
**Rating:** 3
**Confidence:** 4

**Summary:**

This paper explores whether the current pixel-level MLLM is really moving in the right direction. To this end, the authors proposed two new paired benchmarks (PixMMVP and PixCV-Bench) for jointly evaluating VQA and pixel-level grounding. The authors argue that pixel-level MLLM often hurts the model's ability to answer visual questions. They also proposed a simple baseline method (PixFoundation) for extracting good localization segmentation from regular MLLMs that are not trained on this task, and tried to show that MLLMs without pixel-level localization supervision have the potential to outperform pixel-level MLLMs in these tasks.

**Questions:**

1. PixFoundation uses SAM for mask candidate generation and GPT-4o for the critical mask selection. Can you elaborate on the importance of the mask candidate model and mask selection model, and explain whether it is fair to compare this workflow with unified open-source models?

2. Can you provide a more detailed analysis of the computational overhead and inference time of the PixFoundation pipeline compared to the pixel-level MLLMs ? This will help readers understand the trade-offs of the methods in practical applications.

3. For the "PixFoundation" baseline, have you tried using it for pixel-level MLLMs?

4. For future work, do you think it is possible to combine the advantages of supervised pixel-level MLLMs and regular MLLMs, or are they fundamentally in conflict?

**Ethical Concerns:**

["NO or VERY MINOR ethics concerns only"]

**Final Justification:**

Thanks for the author's detailed response. I acknowledge their contributions and proactive engagement, but I maintain that the future path is one of multi-path co-evolution, not a binary choice. The current framing is overly radical and may give a one-sided impression to young scholars.
A more balanced tone would be preferable, especially as SAM itself is a specialized pixel-level model and computer vision has not yet escaped the “suboptimal local minimum.”
Regrettably, the authors also did not address my concern that the performance gaps of pixel-level MLLMs on VQA and referred visual grounding may stem from insufficient relevant training data rather than inherent flaws in the approach.
Thus, I downgrade the rating to "Borderline reject".

**Limitations:**

yes

**Quality:**

3

**Strengths And Weaknesses:**

Strengths:

1. This paper is clearly written, and the figures are very helpful in understanding the main points.
2. This paper proposes two new benchmarks (PixMMVP and PixCV-Bench) that can more comprehensively evaluate VQA and segmentation simultaneously, rather than separately, which is very reasonable and helpful.
3. The experiments compare multiple pixel-level MLLMs with vanilla MLLMs, including quantitative and qualitative analysis, ablation and comparison.
4. The proposed baseline method (PixFoundation) is simple but effective, demonstrating that grounding can be mined from the attention map of a vanilla MLLM without additional supervision.

Weaknesses:

The main weaknesses of this paper lies in its presentation, which may mislead readers who are not familiar with this particular field. The paper only makes a superficial comparison, highlighting the performance differences without digging into the reasons behind them (or at least without providing potential explanations).
This has the potential to create a false opposition between "pixel-level MLLM" and "vanilla MLLM", which may lead readers to prematurely conclude that the former is a flawed or less valuable research direction.
The goal of this paper should be to stimulate discussion on how to improve all types of models, rather than positioning the two methods as adversarial. Specifically:

1. The PixFoundation baseline relies on a strong multi-stage workflow: it uses SAM to generate a large number of high-quality mask candidates and then uses GPT-4o for the final mask selection. Its strong performance is likely due in large part to SAM's excellent multi-granularity mask generation capabilities and GPT-4o's strong visual perception capabilities. This makes the comparison with smaller, independent open source models in Table 1 seem unfair.
This is not a comparison with similar methods because it compares a single end-to-end model to a complex pipeline that decomposes the task into three subtasks: coarse attention localization (via native MLLM), fine mask candidate generation (via SAM), and optimal mask selection (via GPT-4o).

2. In contrast, the results of this paper can also be interpreted more favorably for pixel-level MLLMs. The baseline performance (the M metric) of models such as OMG-LLaVA, GLAMM, and LISA is comparable to or even exceeds PixFoundation Oracle (7B), which fully demonstrates their potential. This shows that a single unified model is able to achieve comparable performance to complex multi-model workflows including SAM and GPT-4o. This suggests that these models are not failures.

3. The performance degradation of some pixel-level MLLMs on VQA and referred visual grounding instruction following may not be an inherent flaw of the pixel-level methods themselves. More likely, they employed a specialized training regimen in which the models were fine-tuned to perform well only on specific task formats and textual cues. This is most likely a problem of insufficient distribution and diversity of the training data rather than a fundamental flaw in the methodology. The current narrative in this paper may be misleading because it does not adequately consider this critical context.

---

> ### Author Rebuttal · Authors · 2025-07-25
>
> We thank the reviewers for their comments. Since the majority of the reviews are on the Reject side, we opt not to respond point by point as a standard rebuttal, but rather provide one response to major points for all the reviewers and provide details on how we plan to improve the work for future submissions
>
> 1- Our work discusses the direction of pixel-level multi-modal large language models, while being end-to-end is favourable indeed, but if it doesn't surpass what we already have and in fact degrades previous capabilities, then we are, of course, heading in the wrong direction. The success of multi-modal large language models was due to pushing the field towards general-purpose models and even better utilizing them in creating embodied agents or user interface control. It was not because we tried so hard to finetune specialized models! So the creation of pixel-level multi-modal large language models for only referring segmentation is an appreciated effort, yet if it degrades previous general-purpose capabilities, then its in the wrong direction. This is judging by how LLMs won over specialized machine translation/sentiment analysis/ or other specialized models tasks in NLP. We don't want to push the Computer Vision field into another suboptimal local minima that we have already gotten out of with MLLMs!
>
> 2- Since our main contribution is the benchmarking efforts, we create baselines that are training-free that show that, without any cumbersome training effort we can still perform visual grounding from vanilla MLLMs. As such, being multistage and using other models like SAM does not impact our message. Our message is, why bother creating new pixel-level models when we already have what we need and with better Q/A and chat abilities! We are not trying to do an apple-to-apple comparison; It is quite difficult to do so with each model having their own architectural choices. Nonetheless, we are trying to ensure that when new models emerge, they surpass what we already have. We are not trying to undermine previous efforts at all, but as standard in the research (at least back in the days), we are only trying to publish what will push the field forward! Aside from that, it is interesting how these apple-to-apple comparison comments did not emerge in OMG-LLava work which was comparing with all methods using older LLMs citing inconsistency in our reviewing system. So, with work like Sa2VA ( Yuan, Haobo, et al. "Sa2va: Marrying sam2 with llava for dense grounded understanding of images and videos." arXiv preprint arXiv:2501.04001 (2025).) using SAM2 will you give it the same comment if you are reviewing it? If no, because it is an integral part of the method design, this is the same case for our baseline too.
>
> 3- On why we need pixel-level segmentation output, beyond region/box-level from a universal view of pixel-level understanding and for certain applications in robotics/remote-sensing/medical image processing, there is still a need for the segmentation task. Not only that when coupled with other pixel-level understanding tasks such as depth estimation and pixel-level tracking, it can prove useful to reinforce each other, and if we found the right design, training recipes, it can reinforce the interaction with language. The majority of the popular MLLMs in 2025 such as Qwen2.5-VL and InternVL3, only provide grounding on the box-level.
>
> 4- We use previous state-of-the-art MLLMs from NeurIPS 2024 (Cambrian-1) and LLava 1.5 variants which should be sufficient to showcase the simple baselines. Moreover, we do show a novel contribution on how we use MLLMs not trained for pixel-level visual grounding as a Judge during the mask selection and we showcase it using GPT-4o and Camrbian-1 itself. More importantly, documenting the observation that grounding in MLLMs can emerge coinciding with other noun phrases that describe parts/location/... of the original referring expression was not documented before. This is not a small insight and can drive a lot of contribution moving forward and in fact one of the holy grails of proper research to have good interpretability beyond just bold numbers. We achieve both!
>
> 5- Regarding our baselines not surpassing the SOA in IoU only, we note that only happened in PixCV-Bench because it is at the end within the distribution of the data used for training these models. It also only happened in LLava 1.5 (7B) vs. GLAMM (7B) where the latter has zero ability to answer questions and is actually using a multiscale decoding scheme, so ofc it would be better in certain scenarios, but its only with 1% difference. We are not going to sacrifice all the chat capabilities for 1% difference. For OMG-LLava (7B), we note that it is using InternLM2, which is by design quite better than LLava 1.5 (Vicuna) and should not be compared apple-to-apple. We just repeat that the purpose of these baselines is to ensure that when we are designing new methods, we are pushing the field forward, improving or maintaining Q/A and chat ability while improving segmentation. We are completely against going back to the era of specialized models, and it is counterintuitive for any researcher working on foundation models.
>
> We plan to improve the work in two major ways:
>
> 1- Include more comparison methods from newer 2025 models, such as Sa2VA
>
> 2- Include analysis on what we call **Grounding Hallucinations**, where models identify the object in the image yet wrongly locate it within the image.
>
> We prefer to leave the work and comments public as such; we do not opt to withdraw the submission since the work was plagiarized while it was being reviewed in a previous venue. Hence, we choose to leave the work publicly available after NeurIPS decisions for its protection.

---

> ### Comment · Reviewer_E95z · 2025-08-07
> **A high-level suggestion**
>
> I feel that the response still takes an adversarial stance. As Reviewer xoMW rightly pointed out, “the goal of the paper should be to stimulate discussion on how to improve all types of models, rather than positioning the two methods as adversarial.” I have a similar comment in my initial review that a more constructive approach would be to identify current limitations across different modeling strategies and propose actionable steps to address them, rather than attempting to prove that a particular direction is wrong.
>
> I fully respect that the authors advocate for a specific direction. However, dismissing alternative approaches, based on the assumption and limited experimental observations that general-purpose models are the “right” path, risks overlooking the diverse opportunities and practical needs of real-world applications. Specialized and end-to-end models, for example, remain highly valuable in scenarios with domain-specific constraints, deployment limitations, or strict performance efficiency requirements. And in some cases, may even outperform general-purpose models on targeted metrics (such as PixCV-Bench). It’s not only difficult but often counterproductive to declare a direction as wrong. A high-level suggestion for the authors is to think their work and contributions within a more inclusive and forward-looking perspective.

---

### Comment · Area_Chair_nCA2 · 2025-08-06
**Re: Rebuttal by authors**

Dear reviewers,

Could you please let us know your feedback based on the rebuttal? Feel free to answer all under the reply for Reviewer xoMW.

Best!
Your friendly AC

---

### Decision · Program_Chairs · 2025-09-17

**Decision:**

Reject

**Comment:**

The paper focuses on the problem of VQA grounding, proposing two pixel-level grounding benchmarks, PixMMVP and PixCV-Bench, as well as a baseline, called PixFoundation, for pixel-level grounding to evaluate.

The paper received four reviews with the following rating: BR - R - R - R. Overall, there was an active discussion with reviewers mainly criticising the confrontative stance of the paper and advising a more nuanced discussion of the topic.

 AC follows the reviewer voting and recommends rejecting the paper in it's current from.